behaviour/ecology

fin whale, *Balaenoptera physalus*, Elephant Island, Antarctica, Scotia Sea, passive acoustic monitoring

**Author for correspondence:**
Elke Burkhardt
e-mail: elke.burkhardt@awi.de

# Seasonal and diel cycles of fin whale acoustic occurrence near Elephant Island, Antarctica

Elke Burkhardt[1], Ilse Van Opzeeland[1,2], Boris Cisewski[3], Ramona Mattmüller[1], Marlene Meister[1], Elena Schall[1], Stefanie Spiesecke[1], Karolin Thomisch[1], Sarah Zwicker[1] and Olaf Boebel[1]

[1]Alfred Wegener Institute Helmholtz Centre for Polar and Marine Research, Am Handelshafen 12, 27570 Bremerhaven, Germany
[2]Helmholtz Institute for Functional Marine Biodiversity (HIFMB), Carl von Ossietzky University, Ammerländer Heerstraße 231, 26129 Oldenburg, Germany
[3]Thünen Institute of Sea Fisheries, Herwigstraße 31, 27572 Bremerhaven, Germany

EB, 0000-0002-5128-4176; IVO, 0000-0001-8369-7234;
BC, 0000-0002-1130-6107; RM, 0000-0002-9896-112X;
ES, 0000-0002-7740-5466; SS, 0000-0002-1105-7710;
KT, 0000-0002-7144-8369; SZ, 0000-0002-1901-8708;
OB, 0000-0002-2259-0035

This study investigates the relevance of the Elephant Island (EI) region for Southern Hemisphere fin whales (*Balaenoptera physalus*) in their annual life cycle. We collected 3 years of passive acoustic recordings (January 2013 to February 2016) northwest of EI to calculate time series of fin whale acoustic indices, daily acoustic occurrence, spectrograms, as well as the abundance of their 20 Hz pulses. Acoustic backscatter strength, sea ice concentration and chlorophyll-a composites provided concurrent environmental information for graphic comparisons. Acoustic interannual, seasonal and diel patterns together with visual information and literature resources were used to define the period of occupancy and to infer potential drivers for their behaviour. Spectral results suggest that these fin whales migrate annually to and from offshore central Chile. Acoustic data and visual information reveal their arrival at EI in December to feed without producing their typical 20 Hz pulse. For all 3 years, acoustic activity commences in February, peaks in May and decreases in August, in phase with the onset of their breeding season. Our results emphasize the importance of EI for fin whales throughout most of the year. Our recommendation is to consider EI for establishing a marine protected area to expedite the recovery of this vulnerable species.

# 1. Introduction

Elephant Island (EI, 61° S 55° W) is the northernmost Island of the South Shetland Island chain near the tip of the Antarctic Peninsula between Bransfield Strait and Shackleton Fracture Zone, in close vicinity to the southern boundary of the Antarctic Circumpolar Current front [1]. Waters around EI are krill-rich [2–5] and part of an important spawning and nursery ground for Antarctic krill (*Euphausia superba*) [6,7]. The area is known for high krill densities [6] with large aggregations occurring mainly north of EI due to the island's bathymetric features [8–10]. These krill stocks probably provide an important and reliable feeding ground for a variety of top predators [11], including baleen whales like humpback whales (*Megaptera novaeangliae*) and, particularly, fin whales (*Balaenoptera physalus*) [10,12,13].

Throughout recent years, increasing numbers of fin whale sightings were reported for the South Shetland Island area [13–19], EI in particular, involving not only an increase in the total number of sightings but also an increase in observed group sizes. Santora *et al.* [10] reported mean group sizes of 11–29 animals from January surveys between 2003 and 2007; Joiris & Dochy [14] noted several groups of up to 20 animals and one of around 100 individuals in March 2012, while Herr *et al.* [20] observed large feeding groups of up to 60 animals in Drake Passage in March 2013. Particularly high aggregations of fin whales were observed off EI [21] during an opportunistic visual survey in autumn 2012. Within 11 days, 116 sightings (cf. electronic supplementary material, table S1) of average group sizes of one to five animals were made, but also larger groups of more than 20 animals and on two occasions, of more than 100 animals, were recorded. During that survey, animals were observed feeding on krill in shallow waters (less than 300 m) and travelling in deeper waters (greater than 1000 m). This increase in recent observations of large fin whale aggregations is particularly noteworthy. While large fin whale aggregations are mentioned in the secondary literature in general terms for highly productive areas [22,23], original reports are lacking for the EI region during the post-whaling period prior to the turn of this century. Hence, while fin whales appear to be slowly recovering after exploitation during last century's commercial whaling (banned in 1976), they are still listed as vulnerable by the IUCN Red List of Threatened Species [24]. Therefore protecting their key habitats should be a primary objective of conservation efforts.

The aforementioned observations pointed our interest towards the general relevance of EI in fin whale ecology, last but not least in light of possible conflicts with fisheries [25] and touristic activities there [26]. Our surmise of fin whales increasingly using the EI area intensively was thereby corroborated by a publication employing passive acoustic monitoring (PAM) at the northwestern shelf break off EI (figure 1, label 'EI-SCRIPPS'), which reported high acoustic activity of fin whales during austral autumn and early winter 2014 [27,29]. However, with these visual and acoustic observations providing only (temporally) selective information (single days for visual and a period of five months only for acoustic data) for the EI region, we opted for long-term PAM to examine fin whale acoustic intensity and occurrence year-round and across multiple years.

While PAM data had been collected a decade earlier for 3 years at a location 192 nautical miles (NM) farther west at the shelf break ([27], and figure 1, label 'WAP'), our study presents the first on-shelf recordings of fin whales in the EI coastal zone (table 1 and figure 1, label 'EI-AWI'). To determine the temporal patterns of fin whale use of the EI shelf and to improve our understanding of this island's role in fin whale ecology, we examined our passive acoustic data for interannual, seasonal and diel patterns and related these graphically to the seasonal patterns of environmental covariates as well as the timing of visual sightings. Environmental covariates included maps of sea ice cover and chlorophyll-a concentrations (Chl-a, as a proxy for phytoplankton concentration) as well as vertical profile data of acoustic backscatter strength (as a proxy for zooplankton and fish presence) at the mooring location. In a more tentative approach, we explored the potential of making meaningful comparisons of the magnitude of call-specific signal-to-noise ratios (SNRs) (so-called fin whale indices) between different recorder deployments (reaching in this case over a decade) to, at least coarsely, capture abundance trends.

# 2. Material and methods

## 2.1. Overview

To obtain information on fin whale acoustic occurrence, passive acoustic data acquisition comprised two separate recordings collected at the same site on the EI shelf (table 1): subsampled AURAL (PAM-AU) data for 38 months from January 2013 to February 2016 and continuous SonoVault (PAM-SV) data for

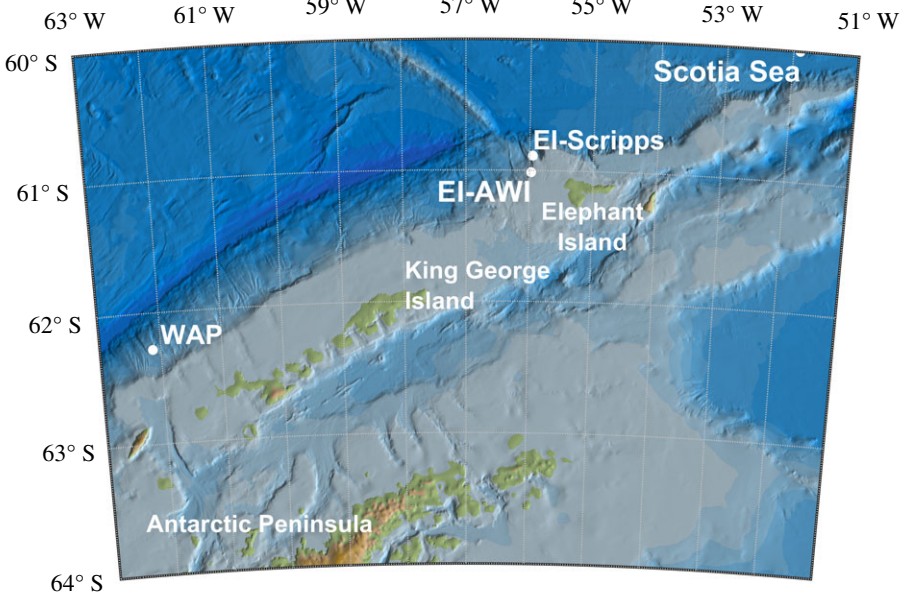

**Figure 1.** Map of mooring locations. White dots indicate the position of mooring AWI 251-1 (EI-AWI) on the Elephant Island shelf, which hosts the passive acoustic recorders analysed in this study, and locations of recorders discussed in Širović *et al.* [27,28] (WAP) and Baumann-Pickering *et al.* [29] (EI-SCRIPPS). Map created using Arndt *et al.* [30,31] and Greene *et al.* [32].

**Table 1.** Geographic settings and recording periods of passive acoustic recorders of this and previous studies.

| study | location (cf. figure 1) | water depth/ recorder depth and location | recorder | recording time start–end duration |
|---|---|---|---|---|
| this study (2021) | EI-AWI 61°00.88′ S 55°58.53′ W | 320 m 210 m and 212 m on shelf edge | AURAL | 16 January 2013–10 February 2016 38 months |
| | | | SonoVault | 16 January 2013–09 Nov 2013 10 months |
| Baumann-Pickering *et al.* [29] | EI-SCRIPPS 60°53.2′ S 55°57.5′ W | 760 m ≈ 10 m above bottom on shelf break | HARP | 5 Mar 2014–17 Jul 2014 5 months |
| Širović *et al.* [27] | WAP (their site S1) 62°16.44′ S 62°10.02′ W | ≈1600 m) ≈ 10 m above bottom on shelf break | ARP | Mar 2001–February 2003 24 months |
| Širović *et al.* [28] | WAP 62°16.69′ S 62°07.80′ W | 1632 m ≈ 10 m above bottom on shelf break | ARP | 28 February 2003– 1 Mar 2004 12 months |

a period of 10 months (January 2013–November 2013). Auxiliary datasets comprised three environmental parameters (table 2): (i) *in situ* active acoustic backscatter data collected between January 2013 and October 2015 to obtain information on prey biomass (zooplankton and fish) at the recording site; (ii) satellite-borne daily sea ice concentration for the entire study period to explore relationships between fin whale acoustic occurrence and sea ice cover; and (iii) satellite-borne monthly Chl-a concentration composites as information for productivity in the wider surroundings of our recording site for comparison with fin whale occurrence.

**Table 2.** Overview of times of data availability. *In situ* point data: Passive acoustic recordings by SonoVault (PAM-SV, continuous at 5333 Hz) and by AURAL (PAM-AU, at 48 kHz subsampled 5 min every hour), as well as acoustic mean volume backscatter strength (MVBS) by moored upward-looking acoustic doppler current profilers (ADCP) (sampled at 76.8 kHz every 3 h 45 min). Mapped satellite-borne data: daily binned sea ice concentration for the full 3 years and monthly composites of chlorophyll a (Chl-a) for full 3 years, with the exception of predominantly cloudy months (May–August). Lighter shadings within a row indicate data being available for only part of the month.

| | 2013 | | | | | | | | | | | | 2014 | | | | | | | | | | | | 2015 | | | | | | | | | | | | | | |
|---|J|F|M|A|M|J|J|A|S|O|N|D|J|F|M|A|M|J|J|A|S|O|N|D|J|F|M|A|M|J|J|A|S|O|N| |D|J|F|
| PAM-AU | | | | | | | | | | | | | | | | | | | | | | | | | | | | | | | | | | | | | | | |
| PAM-SV | | | | | | | | | | | | | | | | | | | | | | | | | | | | | | | | | | | | | | | |
| MVBS | | | | | | | | | | | | | | | | | | | | | | | | | | | | | | | | | | | | | | | |
| Sea ice | | | | | | | | | | | | | | | | | | | | | | | | | | | | | | | | | | | | | | | |
| Chl-a | | | | | | | | | | | | | | | | | | | | | | | | | | | | | | | | | | | | | | | |

## 2.2. Passive acoustic data acquisition

Passive acoustic recordings were collected between January 2013 and February 2016 by two moored autonomous underwater recorders deployed at 61°00.88′ S, 55°58.53′ W, approximately 31 km west-northwest of Minstrel Point, EI, Antarctica (figure 1). Permission to deploy (during Polarstern expedition ANTXXIX/2) and recover (during Polarstern expedition PS96) the mooring in the Antarctic Treaty Area was granted by the German Federal Environmental Agency under permits I 3.5-94003/286 (deployment) and II 2.8-94003-3/347 (recovery). A SonoVault (Develogic GmbH; [33]) and an Autonomous Underwater Recorder for Acoustic Listening (AURAL-M2; Multi-Électronique, e.g. [34]) were hosted by oceanographic mooring AWI 251-1, which was deployed at 320 m water depth with the recorders placed at 210 m (AURAL) and 212 m (SonoVault) depth (electronic supplementary material, figure S1).

The SonoVault recorded continuously at a sampling rate of 5333 Hz with a resolution of 24 bit and data being stored in 10 min long wav-files [33,35]. Due to battery depletion, recording stopped on 9 November 2013, prior to recorder recovery, resulting in a recording period of approximately 10 months (299 days).

The AURAL recorded at 32 768 Hz at a resolution of 16 bit [33]. The recorder was equipped with High Tech Inc. hydrophone HTI-68-MIN with a factory-calibrated sensitivity of $S = -164$ dB re 1 VμPa$^{-1}$. The AURAL's system amplifier was set to $G = 22$ dB with the analogue digital converter's (ADC) peak voltage being 2 V, resulting in a digital gain of $M = 84$ dB. Data were stored lossless in 16-bit wav-files and, based on the saved counts $c$, were converted to sound pressure levels according to

$$\text{SPL [dB re 1 μPa]} = 20 \times \log_{10}(c) - S - G - M = 20 \times \log_{10}(c) + 58.$$

Calibration is based on factory calibration only, no further pre- or post-calibration was performed, nor did we apply any frequency-specific correction of hydrophone sensitivity. According to the manufacturer, the recorder's frequency response is flat within ±1 dB over the usable frequency range.

Due to recorder-specific constraints in battery life and data storage capacities (table 3 for details of recorder specifications), the recorder was set to a duty cycle of 1/12, recording 5 min every hour in order to obtain year-round and multi-year data coverage. Data were stored in 5 min long wav-files. The recorder worked flawlessly throughout the recording period and passive acoustic data were collected for 1120 days.

## 2.3. Passive acoustic data analysis

Fin whales produce a variety of short low-frequency, high-intensity vocalizations [36–38]. The most prominent call is the so-called 20-Hz pulse [39] (figure 2), which has been recorded throughout oceans worldwide [27,38–42]. The 20 Hz pulses are emitted in regular sequences, creating a stereotyped form of song assumed to be produced only by males [39,43], or in irregular sequences possibly used when socializing [39,44]. In the Southern Ocean, the 20 Hz pulse typically sweeps from 28 Hz down to 15 Hz, lasting approximately 1 s [27]. In some regions, the 20-Hz pulse is accompanied by a higher frequency component (in Antarctic waters around 85 to 89 Hz [27,29] or near 99 Hz [28,45,46], depending on region) which might serve as a differentiation feature for fin whale populations [45]. Fin whales also produce frequency-modulated downsweeps, which generally range between 100 and 30 Hz and are often referred to as 40-Hz calls [36,47]. This call has been attributed to calling during social interaction and foraging, but it is less well described in the literature than the 20-Hz pulse.

**Table 3.** Specifics of recordings and metrics used in PAM data analysis.

| metric | DAO | LTS, FIN21, FIN86 | CAB |
|---|---|---|---|
| recorder type | AURAL | AURAL | SonoVault |
| sampling rate [Hz] | $2^{15}$ (32 768) | $2^{15}$ (32 768) | 5333 |
| decimation to [Hz] | none | none | 500 |
| FFT length [points] | 20 000 | $2^{14}$ (16 384) | 334 |
| FFT length [s] | 0.61 | 0.5 | 1.5 s |
| filter | Hanning | Hamming | Hanning |
| overlap | 50% | 50% | 90% |
| FFT frequency resolution | 1.6 Hz | 2.0 Hz | 0.7 Hz |
| duty cycle | 5 min/60 min | 5 min/60 min | 60 min/60 min |
| subsampling screening | all files on every second day | every second file on all days | every seventh file on every sixth day |
| total no. files screened | all | all | 1027 (219 NA) |

This study equates the occurrence of 20-Hz pulses to the presence of fin whales within the recorder's acoustic range (cf. sections on Sound propagation). No distinction is being made between song and non-song vocalizations as 20-Hz pulses were at times exceedingly numerous and intense, precluding a detailed call structure investigation. The simultaneous higher frequency component in the 85–89 Hz range also occurs regularly in our recordings, averaging at $85.6 \pm 1.5$ Hz ($\pm$ 3 dB band width) for the 10 min spectrogram partly shown in figure 2 and $86 \pm 4$ Hz ($\pm$ 3 dB band width again) when averaging over the full long-term spectrogram (LTS-AU) (featuring a 2 Hz resolution only). This HF-component serves as supporting evidence for fin whale presence if the low-frequency signal in the 20 Hz region overlapped with Antarctic blue whale (*Balaenoptera musculus intermedia*) Z-calls.

Using these pulses at 20 and 86 Hz as indication for fin whale acoustic presence, this study uses four types of data/analysis combinations, which provide different metrics of whale presence with varying levels of temporal resolution and coverage.

— **LTS-AU:** A *long-term spectrogram* of 38 months of *AURAL* data (figure 3*a*).
— **FIN21-AU and FIN86-AU:** *Fin whale index* of *AURAL* data, calculated as the SNR for the 21 and 86 Hz signal bands and their adjacent noise bands, respectively (figure 3*c*).
— **DAO-AU:** Examination of *AURAL* data for *daily acoustic occurrence* of fin whales (figure 3*d*, grey shading).
— **CAB-SV**: Call abundance of fin whale 20-Hz pulses from 10 months SonoVault data.

Please note the explicit use of term 'acoustic occurrence', reflecting on what was recorded and detected in the PAM data, rather than 'acoustic presence', which describes a vocalization to have happened. This distinction might appear petty here, but will become relevant when discussing the odds of meaningful comparisons of FINs between different recorders, as acoustic presence might not necessarily result in acoustic occurrence depending on background noise levels or sound propagation.

Likewise, we explicitly seek to distinguish the term 'call abundance', which we here use to represent the number of calls discernible in a recording from the term 'call rate', which commonly is understood as the frequency an animal repeats a certain call. Call rates and call abundances are linked in a non-trivial way by the number of animals calling, sound propagation and recorder characteristics. Hence call rate is an emissive attribute concerning a single individual (emitted calls per time unit), whereas call abundance is an immission metric, concerning the entire audible population (recorded calls per time unit).

### 2.3.1. Long-term spectrograms (LTS-AU)

Spectrograms (figure 2) visualize the temporal evolution of sound spectra by mapping the spectral densities to colour or grey space and plotting the acoustic energy present per frequency band (usually 1 Hz) and time step. Minimizing compromising effects of mooring noise, an LTS covering the full

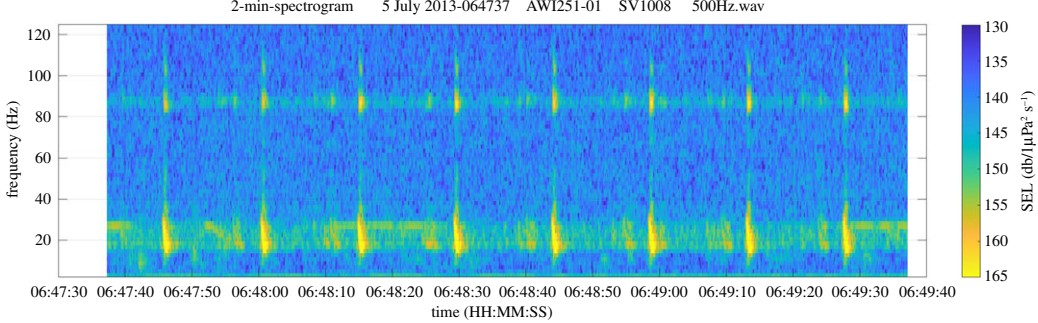

**Figure 2.** Spectrogram of a hydroacoustic recording off EI (SonoVault 5 July 2013 06.45.37, resampled at 500 Hz): eight fin whale 20-Hz pulses with accompanying 86 Hz component directly above.

recording period (38 months, figure 3*a*) was composed from power spectral densities (PSD) of the quietest period of 10 s within every second file of 5 min duration (i.e. the file length for the AURAL data). PSD were calculated after Welch [48] using a window length of 2 s with 50% overlap and a fast Fourier transform size of $2^{14}$ data points (table 3). Using the quietest 10 s minimizes the impact of periods dominated by mooring-induced noise (like flow or shackle noise which is particularly notable in this shelf-based deployment), which could negatively affect detection of the transient fin whale pulses.

### 2.3.2. Band acoustic power (FIN21-AU and FIN86-AU)

Acoustic power present at 21 and 86 Hz was calculated as a proxy for fin whale acoustic occurrence. Analysis was performed on AURAL data for every second 5 min file independently based on data drawn from the aforementioned LTS. Results were filtered by applying a 14-day running mean, representing the typical time scale at which environmental changes (like substantial changes in sea ice concentration, insolation and biomass response) occur. Acoustic power is presented as SNR (figure 3*c*) of the 86 Hz fin whale call band versus two bracing noise bands at 80–82 and 92–94 Hz, in analogy to the approach taken by Širović *et al.* [27,28]

$$\text{SNR} = \frac{S_{(86-88\text{Hz})}}{0.5 \times [S_{(80-82\text{Hz})} + S_{(92-94\text{Hz})}]},$$

with $S$ representing the power spectral density.

Similar calculations were performed for the low-frequency pulses (20–22 Hz signal band and 8–10 and 30–32 Hz noise bands); however, these proved to be less representative of fin whale presence due to the presence of Antarctic blue whale Z-calls overlapping with the 20–22 Hz signal band [28]. Please note that bands for FIN calculations have not yet been standardized between studies, as desirable when making comparisons between different recorders, yet it might also be difficult to do so, when call frequencies shift with time [49]. Table 4 gives an overview of definitions as used by studies cited in this manuscript. Note that slightly different FIN17 and FIN89 indices are used by other authors [27,50].

### 2.3.3. Daily acoustic occurrence (DAO-AU)

Screening for daily acoustic occurrence involved spectrographic and/or aural inspection of the full multi-year (i.e. 2013–16) AURAL data at 2-day resolution using Raven Pro 1.5 (Bioacoustics Research Program 2014, Cornell Lab of Ornithology, Ithaca, NY) with spectrogram settings (see [54] for details) kept constant across analyses, except for brightness and contrast being adjusted when necessary. For a given day, data were visually screened up to the moment the first pulse of the respective day was detected.

During the screening, only sounds that were clearly recognizable as individual signal were logged. However, 20-Hz pulses produced by entities of distant fin whales may form a continuous 'chorus' band in the frequency range between 15 and 30 Hz [28,55]. These bands were not considered as a proxy of acoustic occurrence of fin whales during the DAO analyses (yet by the SNR metric, see section on FIN21-AU), to focus this metric on nearby calls and local habitat use. DAO-AU hence emphasizes the presence of solitary fin whales or those closer to the recorder, while FIN21-AU and LTS-AU also include the chorus of more distant, indistinguishable callers.

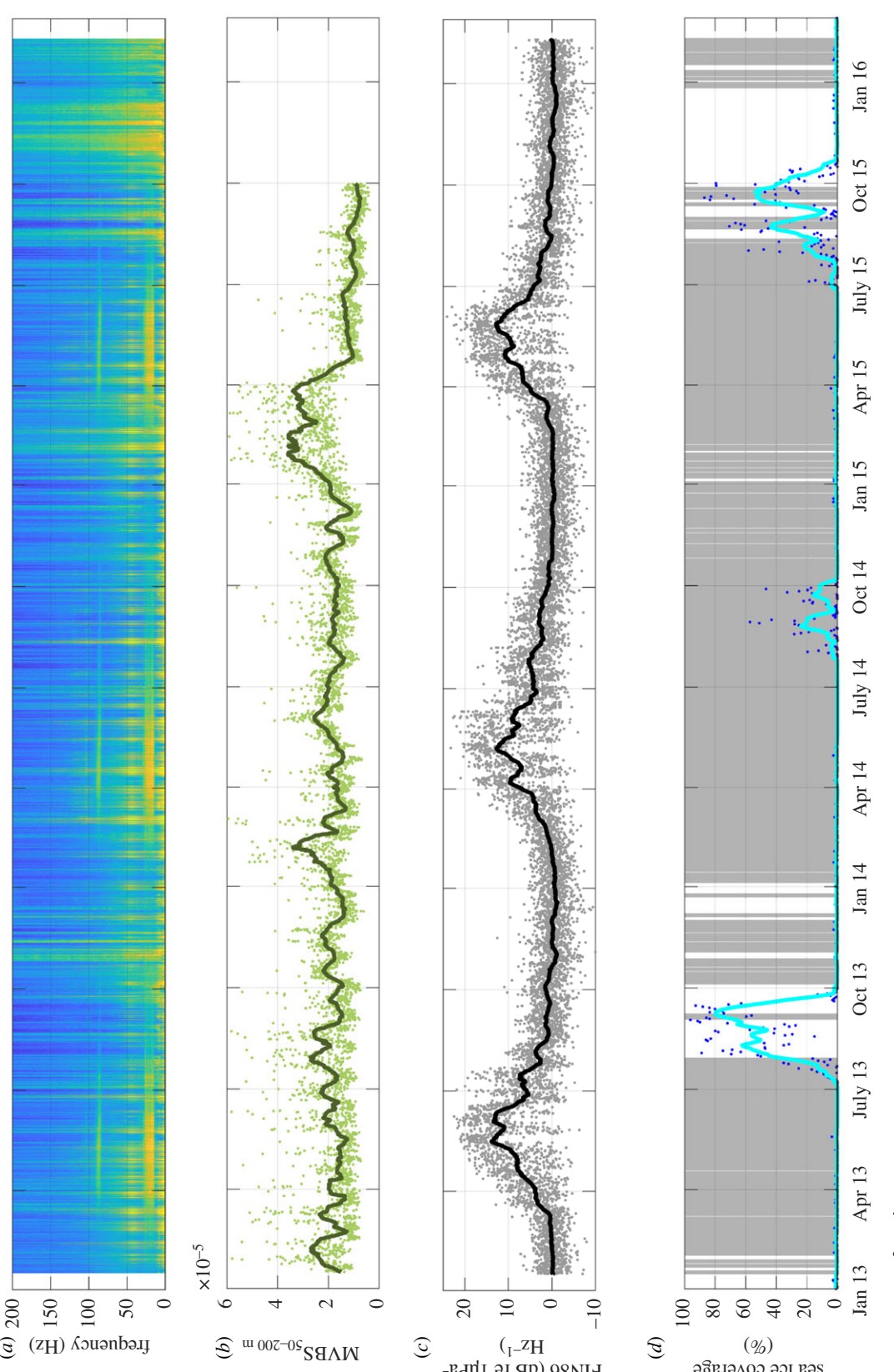

**Figure 3.** (*a*) LTS [dB re 1 µPa² Hz⁻¹] of AURAL recordings from January 2013 to February 2016; (*b*) dots: MVBS [dB] of upward-looking ADCP; line: 14-day arithmetic running mean. (*c*) FIN86 index, i.e. the signal-to-noise ratio (SNR [dB re 1 µPa² Hz⁻¹]) at 86 Hz; (*d*) DAO (daily acoustic occurrence, grey bars) of fin whales and sea ice coverage [%] (blue dots and cyan line).

**Table 4.** Overview of FIN bands as used in the literature and herein. Please note that 1 Hz wide bands have usually not been specified explicitly by the authors, i.e. some ambiguity exists, whether an x Hz band refers to the x Hz or to x + 1 Hz band (most likely it does and is assumed in this table) or to the x − 0.5 to x + 0.5 Hz band.

| study | region | lower adjacent noise band [Hz] | calling band [Hz] | upper adjacent noise band [Hz] |
|---|---|---|---|---|
| **this study** | **Antarctic** | **8–10** | **20–22** | **30–32** |
| Baumann-Pickering et al. [29] | Antarctic | 9–10 | 21–22 | 33–34 |
| Buchan et al. [50] | Juan Fernandez Archipelago | 10–11 | 16–22 | 35–36 |
| Simon et al. [51] | Arctic | 13–17 | 19–28 | 33–37 |
| Nieukirk et al. [52] | North Atlantic | 8–13 | 19–22 | 40–45 |
| Širović et al. [53] | Southern California Bight | 10–11 | 22–23 | 34–35 |
| **this study** | **Antarctic** | **80–82** | **86–88** | **92–94** |
| Širović et al. [28,29] | Antarctic | 80–81 | 89–90 | 98–99 |
| Buchan et al. [50] | Juan Fernandez Archipelago | 70–71 | 84–86 | 100–101 |

### 2.3.4. Call abundance (CAB-SV)

SonoVault's continuous recordings were analysed for call abundances (CAB), implementing an increased daily resolution (10 min every 70 min) at the expense of a reduced weekly resolution when compared with the analyses of AU data. Using acoustic data decimated to 500 Hz, spectrograms were calculated using RAVEN 1.4 (Hanning-window, FFT 334 points, overlap 90%, frequency resolution 0.75 Hz, time resolution 1.33 s, displayed frequency range (linear) 0–125 Hz). Screening for 20-Hz pulses was performed audio-visually; applying a subsampling scheme as suggested in Harris et al. [56] to manage temporal constraints. Analysis was implemented as counting the number of 20-Hz pulses within a 10 min file (further details in Mattmueller [57]) from every seventh 10 min file of every sixth day. This resulted in a total of 1028 available 10 min files for the recording period 16 January 2013 to 9 November 2013, consistently covering all months of 2013 except for December, for which no recordings exist. Of the available files, a total of 219 files (mainly August, September and October) had to be excluded from the analysis due to excessive ambient noise. Only clearly discernible calls were counted and chorus bands were not considered as 'calls'.

During late-summer/early-autumn, call numbers peaked to up to 150 calls per 10 min file. However, as call abundances become rather speculative for files with values greater than 100 calls (as calls start to merge into another) or for files with continuous high energy in the corresponding energy band (from e.g. abiotic noise caused by storms), such files were classified as 'greater than 100' without further quantification.

Diel calling pattern analysis was performed on CAB-SV data from March–July, as these are the months with high 20-Hz pulse detections and a clear solar day/night pattern. To test for statistically significant differences in call activity in dependence on light period a Kruskal–Wallis test (as a Kolmogorov–Smirnov test showed data not to be normally distributed, $p < 0.001$) was applied and differences were accepted as statistically significant, when $\alpha < 0.05$.

## 2.4. Environmental data

### 2.4.1. Acoustic backscatter

Presuming that zooplankton are the main scatterers of sound in the frequency range of tens to hundreds of kHz (e.g. [58–66]), acoustic Doppler current profilers (ADCP) have been proven to be a useful tool to describe spatial and temporal patterns in the distribution of zooplankton biomass and its diel vertical migration in many oceanic regions [67]. Likewise, this study made opportunistic use of data collected by an ADCP (Teledyne RDI Workhorse Longranger) included with our mooring for (physical) oceanographic research, i.e. velocity profiles, to obtain mean volume backscatter strength (MVBS). The

instrument featured a four-beam, convex configuration with a beam angle of 20° and a frequency of 76.8 kHz, which corresponds to an acoustic wavelength of approx. 2 cm, i.e. the size class of euphausiids and small fish [65]. The instrument was moored at a nominal depth of 314 m in upward-looking mode and measured horizontal and vertical currents and acoustic backscatter intensity from 16 January 2013 until 30 September 2015. Heading, pitch, roll and temperature data were also collected. Bin size was set at 16 m (exception first bin 24.52 m). During this period, 6330 ensembles of 45 pings at 5 min ping interval were collected, resulting in a temporal resolution of 3 h 45 min. While Cisewski *et al.* [66] optimized their instrument set-up towards resolving the vertical migration of e.g. krill, the (oceanographic) set-up used here, particularly the temporal resolution, only allows a rather coarse analysis for potential diel cycles.

The ADCP recorded echo intensity on a 0 to 255 automatic gain control count scale. The echo intensity $E$ was converted to the MVBS (dB) after the version of the sonar equation presented by Deines [68]

$$\text{MVBS} = C + 10\log_{10}\left[(T_x + 273.16)R^2\right] - L_{DBM} - P_{DBW} + 2\alpha R + K_C(E - E_r),$$

where $C$ is a system constant delivered by the manufacturer (which includes transducer and system noise characteristics and is −159.1 dB for the Workhorse Longranger), $T_x$ is the temperature of the transducer (°C), $R$ is the range along the beam to scatterers (m), $L_{DBM}$ is the $10\log_{10}$ transmit pulse length (m), $P_{DBW}$ is the $10\log_{10}$ transmit power (W), $\alpha$ is the sound absorption coefficient of seawater (dB m⁻¹), $K_c$ is a beam-specific scaling factor and $E_r$ its real-time reference level. MVBS was calculated per ensemble [68] for the 50–200 m range. (Due to ADCP side lobe effects, air bubbles and sea ice, the MVBS data in the uppermost 50 m are questionable and are excluded from our analysis.) A 14-day running average was applied to the ensemble data (figure 3b). To examine for the potential existence of diel cycles in backscatter depths, monthly mean noon and midnight depths of the backscatter maximum and their difference (i.e. migration amplitude) were estimated by selecting the backscatter maximum depths collected between 10.00 and 14.00 and between 22.00 and 02.00 (next day) local time, respectively.

### 2.4.2. Sea ice concentration

Sea ice concentration (%) in the EI region was obtained from daily satellite data sea ice concentration processed and provided by the University of Bremen[1] [69] at a resolution of 6.25 × 6.25 km (electronic supplementary material, figure S2). Spatially averaged, daily ice concentration was calculated for all sea ice pixels located within a 30 km radius around the recording site, representing an area of $2.8 \times 10^3$ km². We consider the sea ice conditions in this area to be representative of the average sea ice conditions experienced by the fin whales recorded by the PAM units as the assumed 30 km radius mirrors the acoustic detection range of fin whale 20-Hz pulses with received levels[2] (RLs) RL greater than 90 dB re 1 µPa², i.e. louder than the typical (frequency band-specific) background noise level (cf. electronic supplementary material, figure S3). The radius was determined by acoustic propagation modelling (cf. §§2.4.4 and 3.2.4 below). Fourteen-day running averages of spatially averaged sea ice concentration were calculated for comparison with similarly processed data on fin whale acoustic occurrence (figure 3d, blue line).

### 2.4.3. Chlorophyll-a

To investigate the development and decay of phytoplankton blooms in the study region during the sampling period, maps of the near-surface chlorophyll-a concentration derived from satellite remote sensing were used. Satellite-sensed surface chlorophyll distributions were acquired using Ocean Colour Climate Change Initiative dataset, v. [3.1], European Space Agency, available at http://www.esa-oceancolour-cci.org/ for the period January 2013–February 2016.

### 2.4.4. Sound propagation

Sound propagation was modelled in three dimensions using the dBSea Software (v. 2.2.5, developed by Marshall Day Acoustics and Irwin Carr Consulting, UK). A constant sound velocity profile was assumed, derived from the CTD profile taken by an Argo float (AWI 0246, WMO 7900409) in summer on

---

[1]https://seaice.uni-bremen.de/sea-ice-concentration/amsre-amsr2/.

[2]Reference value 1 µPa² according to ISO 18405 3.2.1.1 Note 3.

16 January 2013 at 60°02.142′ S 57°29.25′ W, and in winter on 12 October 2013 at 57°50.658′ S 39°13.53′ W, i.e. northwest of EI in the deep waters of the Scotia Sea, in the direction of where fin whales are expected in wintertime.

The sound propagation was modelled for a calculation grid of 100 points for x- and y-direction each and of 500 points for z-direction (resulting in step sizes of 2248 m in x-direction/longitudinal, 2282 m in y-direction/latitudinal and 10.5 m in depth). A source solution of 100 radial slices and 500 range points was chosen resulting in 3.6° slice step angle and 452 m range step.

The propagation model solved for normal modes, employing a silt bottom. While sediment data was unavailable for the EI shelf and slope, we selected silt as sediment type which has been documented at some distance to the east [70]. The software's default settings for this sediment type were used, i.e. sediment sound velocity of 1575 m s$^{-1}$, density of 1700 kg m$^{-3}$ and attenuation of 1 dB per wavelength [71]).

A virtual source of level[3] $SL = 180$ dB re. 1 µPa$^2$ m$^2$ (third-octave band from 18 to 22 Hz, centred at 20 Hz) was chosen based on the average of source levels reported for fin whale calls globally [36,39,72–75]. The source was placed at a depth of 210 m (i.e. the recorder's deployment depth), while RLs were displayed for 21 m, representing the model's grid point (see next paragraph) closest to an assumed (virtual) fin whale calling depth of 20 m [76]. These choices represent the inverse presumed acoustic path of a whale vocalizing at *ca* 20 m and the recorder recording at 210 m depth.

### 2.4.5. Insolation

To test if fluctuations in call abundance were related to changing light conditions, three light periods were introduced: 'light', 'twilight' and 'dark' [47,77,78]. The period after sunrise and before sunset was defined as 'daytime' period, more specifically hours with a sun altitude greater than 0°. 'Twilight', refers to the period with a sun altitude between 0° and −12° (nautical twilight), both during dusk and dawn. The period between the end and start of nautical twilight, with a sun altitude less than −12° was referred to as 'night-time'. Data for sunrise, sunset and nautical twilight were obtained from the US Naval Observatory sunrise/sunset table and nautical twilight table. Local time at EI was set to UTC − 3 h 40 min, i.e. the true local time for the longitude of 56°W, rather than the time zone UTC − 4 h.

## 3. Results

### 3.1. Acoustic data results

#### 3.1.1. Long-term spectrogram (LTS-AU)

The 3-year LTS of the AURAL recordings (figure 3a) shows the energy bands caused by the fin whale 20-Hz pulses and their 86 Hz component with most energy occurring between mid-March to mid-July of each year (orange/yellow horizontal lines at the respective frequency). This pattern is recurrent for each of the three consecutive years with AURAL recordings. Manual screening (see below) of the 20–30 Hz band, however, revealed that it comprises not only fin whale 20-Hz pulses, but also Antarctic blue whale Z-calls, a potentially significant contamination when using this band as a proxy for fin whale acoustic occurrence. The LTS also reveals regularly occurring, energetic broadband events (vertical yellowish stripes in figure 3a), which again require caution when analysing and interpreting 20 or 86 Hz power spectral levels. These events most likely are caused by storms and possibly mooring flow noise during peak tidal flow.

Besides fin whale vocalizations, the frequency range of EI recordings analysed here also contains acoustic signatures of the following species: Antarctic blue whale, Antarctic minke (*Balaenoptera bonaerensis*) whale as well as crabeater (*Lobodon carcinophaga*) and leopard seals (*Hydrurga leptonyx*) [54]. None of these acoustic signatures overlap in frequency space with the fin whale 20-Hz pulses, except for the Antarctic blue whale Z-call. However, Z-calls exhibit bi-tonal characteristics, whereas the 20 Hz pulse is a pulsed downsweep, which allows for distinguishing the two reliably during manual screening.

---

[3]Reference value 1 µPa$^2$ m$^2$ according to ISO 18405, 3.3.2.1, Note 3.

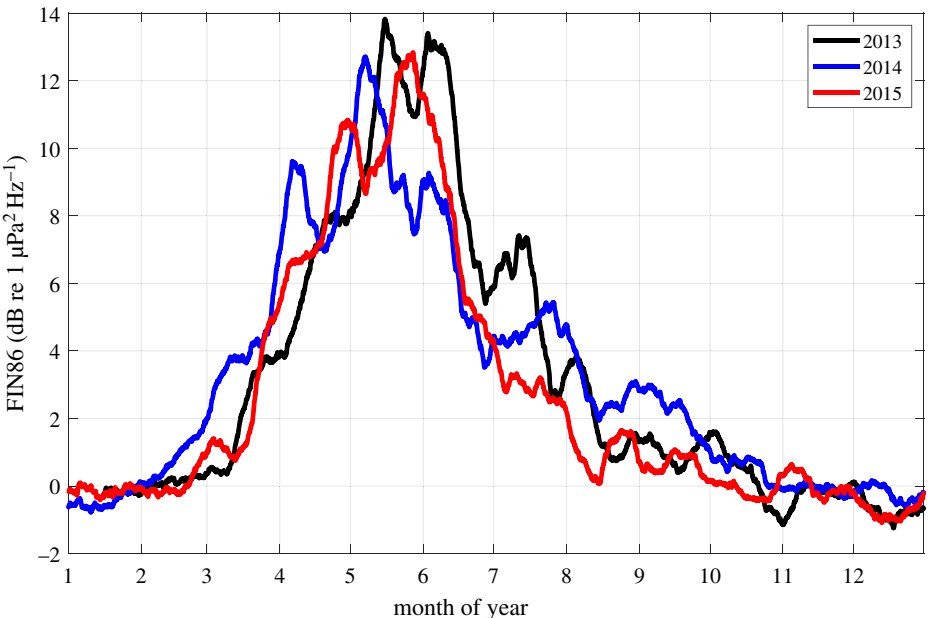

**Figure 4.** Ratio of acoustic power (i.e. SNR) for the 86–88 Hz band versus the arithmetic mean of the 80–82 and 92–94 Hz bands. Two-hourly data were subjected to a 14-day arithmetic running mean.

### 3.1.2. Band acoustic power (FIN21-AU and FIN86-AU)

While the LTS provides a qualitative overview of the entire dataset, including the signal of interest and its possible contaminations, a quantitative understanding is attained by examining spectral SNRs. SNRs of the 3-year AURAL data for the 86 Hz fin whale band (FIN86-AU, figure 3*c*) show that fin whale acoustic power waxes and wanes in a similar manner for the 3 years. For all years, FIN86-AU increases throughout March and April, peaks in May at around 13 dB and decreases from June to September, before vanishing from October to February (figure 4). A similar progression is observed for the FIN21-AU (electronic supplementary material, figure S4), given some additional uncertainties from the overlapping blue whale Z-call. A much smaller, secondary peak occurs during the second part of July in the 86 Hz band only for 2013.

Interannual variation in the timing of the peak power are notable for both bands. In 2014, both waxing and waning of acoustic power precedes those of 2013 and 2015 by around a fortnight, however, with considerable variability between one and three weeks. Local maxima occurred between April (2014) and mid-June (2013) with minor variations (about 1 dB) in peak power, with 2013 being the most intense and 2014 and 2015 slightly less intense years.

### 3.1.3. Daily acoustic occurrence (DAO-AU)

Manual screening of the AURAL data for fin whale daily acoustic occurrence (DAO-AU, figure 3*d*, grey shading) showed that fin whales may occur acoustically in any month of the year. Fin whale calls were identified during the majority of days throughout the year, amounting to 65%, 93% and 61% of days for 2013, 2014 and 2015, respectively. During periods of increased ice cover, fin whales were generally acoustically occurring during fewer days.

### 3.1.4. Call abundance (CAB-SV)

Changes in SNRs as observed in FIN21-AU and FIN86-AU may be induced by several biotic factors: increasing number of callers, callers changing distance to the recorder, or callers changing call rates, call intensity or calling depth. Additionally, variations in sound propagation might be a confounding factor. To explore which of these might have caused the observed waxing and waning of FIN21-AU and FIN86-AU, we examined call abundances from SonoVault data (CAB-SV).

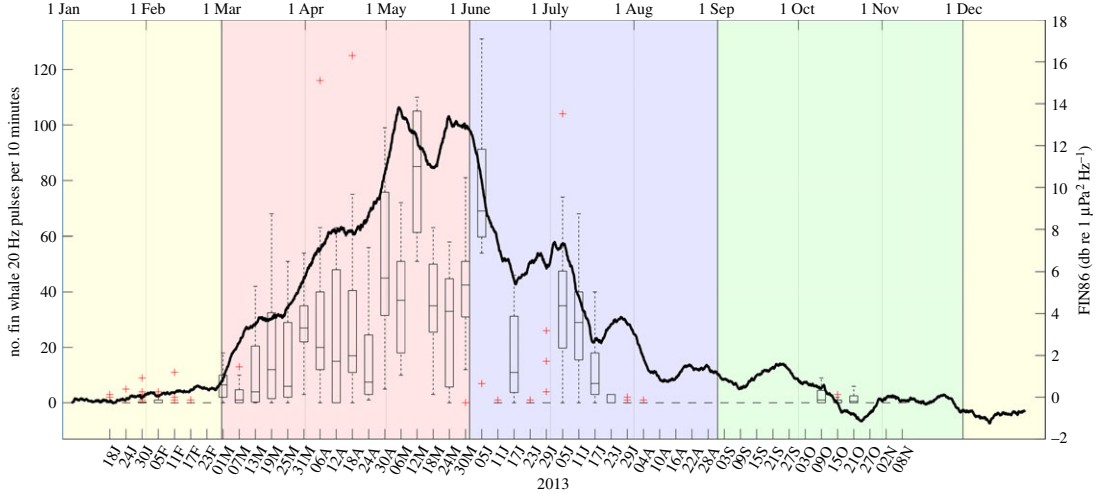

**Figure 5.** Black line: Time series of FIN86-AU index [dB re 1 µPa² Hz⁻¹]; Boxplot: number of fin whale 20-Hz pulses per 10 min file (i.e. 10·CAB-SV) for each day analysed (indicated by lower x-axis labels). For a given day, boxes represent up to 21 10 min files, depending on how many 10 min files were recorded during that day and how many had to be discarded due to excessive chorus or background noise. Boxes indicate the 25th and 75th percentile range, respectively, and lines indicate median values. Red 'pluses' represent outliers. Upper axis: running date.

### 3.1.4.1. Seasonal pattern

Call abundance (CAB-SV) gave a 11 910 total of 20-Hz pulses for the overall recording period between 18 January and 8 November 2013 (figure 5, boxplot). These data show that the increase in SNR in both fin whale bands from March to July (figure 5, black line) can in large parts be explained by an increased number of calls, instead of increased call amplitude: while call abundances increase from less than 1 min⁻¹ to around 10 min⁻¹, i.e. by a factor of 10, SNR increases by about 10 dB, which matches the formal relationship Level($N_2/N_1$) = 10 log$_{10}$($N_2/N_1$) with N being the number of callers (acoustic sources).

Most (99% of all calls) fin whale calls occurred between 1 March and 17 July 2013, with the median peaking on 12 May 2013 at 84 calls per 10 min count interval (range 58–131 calls/10 min). Within 1000 min of notably high call activity during five analysed days in May 2013, a total of 4301 calls were logged, resulting in CAB = 258 h⁻¹. In August (total 1 call) and October (total 61 calls), fin whale calls occurred only sporadically. Calling becomes more regular in March with more count intervals containing calls and higher counts per interval, peaking in the first half of May (greater than 100 calls per 10 min file). A slight decrease in call activity occurs during June followed by a secondary short period of increased calling in July which is then followed by a rapid decrease in CAB-SV, resulting in a gap of acoustic occurrence from late July until the beginning of October. Note that the peak time and the mentioned short period of increased CAB-SV is in accordance with the timing of the peak and the shoulder peak as observed in the FIN21-AU and FIN86-AU (figure 5, black line).

The CAB-SV SonoVault data (figure 6) also exhibited fin whale calls during all months of the recording period (January to November 2013), except for September and November 2013. However, only one call was registered in August 2013. Twenty-Hertz pulses occurred for 9 of 11 months in the SonoVault data (CAB-SV): January to August but not in September and November, with December not sampled.

### 3.1.4.2. Diel calling patterns

SonoVault call abundances (CAB-SV) between January and November 2013 appear to lack a diel cycle in relation to the local light regime (figure 6). This (visual) impression is corroborated by applying a Kruskal–Wallis test for the null-hypothesis 'do average call abundances (CAB) vary between light-regimes'. A positive (statistically significant) test of this hypothesis could be considered an indication of the calls being feeding related, as krill is (presumably) migrating vertically during the day, resulting in differences in food availability (i.e. occurring in denser patches at greater depths during daytime and moving up towards the surface during night). Light regimes coded as three ordinal

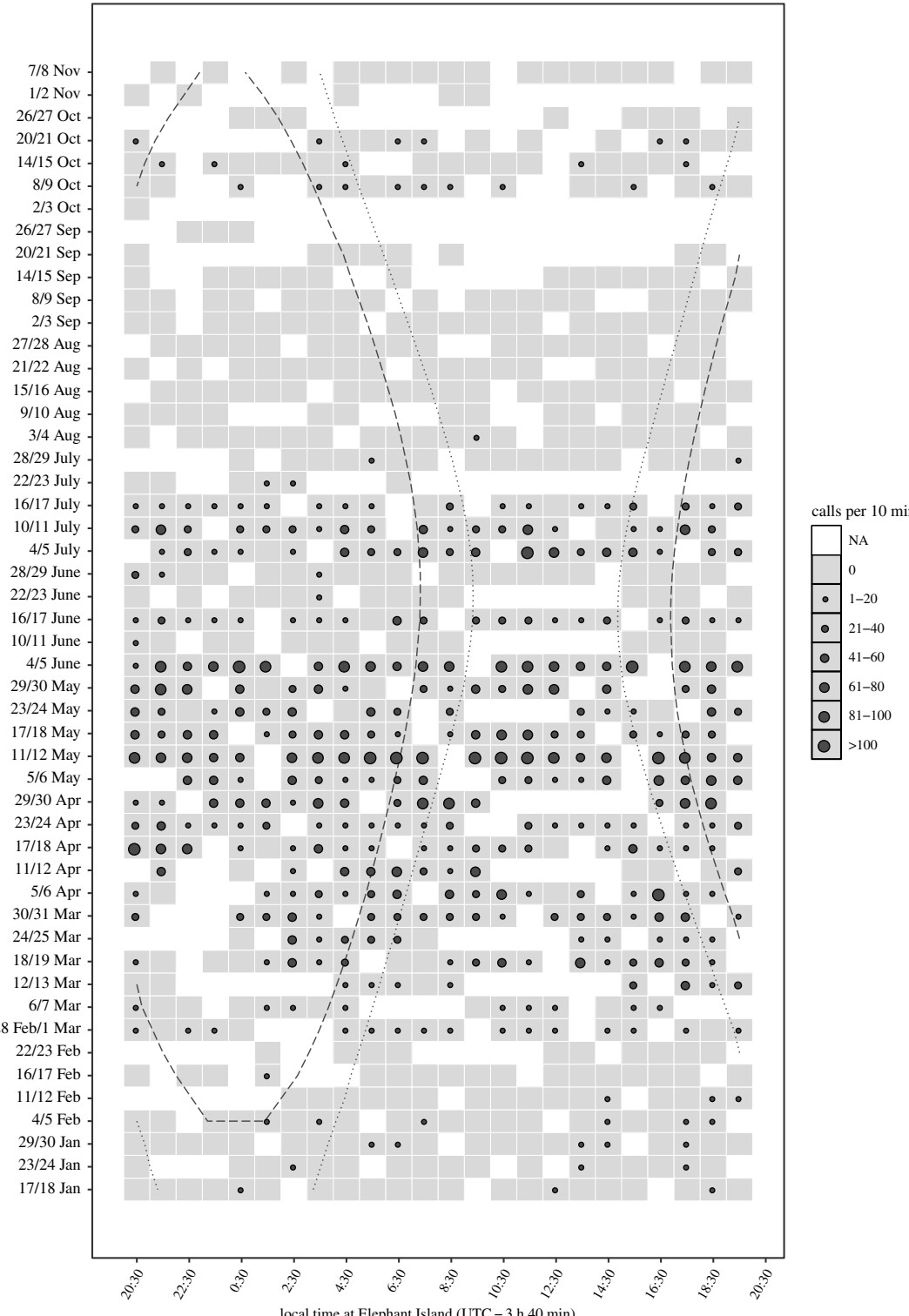

**Figure 6.** Diel calling pattern of fin whale CAB (per 10 min) between January and November 2013 near EI. The x-axis shows the local time at EI (UTC−3 h 40 min), the y-axis shows the days analysed. Grey squares indicate manually analysed files, white squares indicate files not analysed for technical reasons (e.g. excessive background noise). Black dots indicate classes of CAB (per 10 min file) by size. Grey squares without a black dot indicate files CAB = 0. Viewed from noon, the grey dotted line indicates the transition from daylight to twilight period, the dashed line the transition from twilight to night-time.

variables (groups) representing 'light', 'twilight' and 'dark' result in 2 degrees of freedom. With each of the three groups comprising more than nine samples (figure 6), a chi-squared distribution may safely be assumed, indicating (for d.f. = 2 and $\alpha$ = 0.05) a critical value of 5.99. This value is far higher than Chi$^2$ =

**Table 5.** Timing of peaks in ADCP MVBS and FIN86-AU.

| year | figure 3 | 2013 | 2014 | 2015 |
| --- | --- | --- | --- | --- |
| ADCP MVBS | b | January–February | January–February | January–Mar |
| FIN86-AU | c | May | May | May |
| sea ice concentration | d | Aug–Sep | Aug–Sep | Aug–Sep |

1.86 obtained for this dataset ($p = 0.39$), indicating that any difference in CAB between light regimes is insignificant.

## 3.2. Environmental data results

### 3.2.1. Acoustic backscatter

Figure 3b displays the ADCP MVBS calculated for the 50–300 m water column. The values are understood as a proxy for fin whale prey availability, i.e. krill or fish. Elevated MVBS values occurred in February 2014 and throughout February and March 2015, with only a local, smaller peak observed in February 2013. Generally, it is observed that MVBS peaks unequivocally occur prior to peaks in FIN86-AU, which again clearly occur prior to peaks in sea ice concentration, figure 3 and table 5.

Noon and midnight maximum backscatter depths reveal a somewhat unexpected pattern (cf. electronic supplementary material, figure S5). Monthly averaged noon values vary between around 60 to 80 m from August to April (with the notable exception of November), while values are greater (i.e. deeper, around 140 to 160 m) between May and July. Contrastingly, midnight values vary around 40 to 80 m throughout the year, except for October November, when the depth of maximum backscatter strength is around 120 m. The difference between the two, i.e. the amplitude of the daily migration, indicates a pronounced upward vertical migration in May–July, whereas only an order of 20 m displacement is observed for the remainder of the year (with the exception of October, which exhibits an inversion of the pattern, i.e. midday values being shallower than midnight values).

### 3.2.2. Sea ice concentration

While the area around the recording site is free of sea ice throughout most of the year, considerable interannual variability of the average sea ice concentration exists during austral winter (figure 3, cyan line). In 2013, ice concentration near the mooring location exceeded 60% during much of August and September, whereas for the other years, ice cover was substantially lower (for 2014 barely exceeding 20%) and seemed to fluctuate more during this period.

Sea ice occurs mostly to the southeast of EI (cf. electronic supplementary material, figure S2), i.e. an area with reduced acoustic propagation ranges towards the recording moorings. With the exception of August 2013 and September 2013 and 2015, the primary acoustic range to the northwest of the recorder is all but ice free for the entire observational period (January 2013–December 2015).

### 3.2.3. Chlorophyll-a

By and large, no satellite-borne chlorophyll-a data exists for the periods May to August, i.e. the period during which most fin whale calls were recorded, presumably due to wintertime cloud and sea ice cover. However, Chl-a values derived for the period from October to March, i.e. spring to summer, indicate the development of an algal bloom, as noticeable by an order of 10-fold increase in Chl-a concentration from minimum values around 0.1 mg m$^{-3}$ to greater than 1 mg m$^{-3}$, with highest values occurring between January and March (electronic supplementary material, figure S6). This is in line with the general development of blooms in the southern Scotia Sea thought to be governed by the timing and pattern of sea ice retreat during spring [79]. However, the vanishing of sea ice around the mooring in October occurs three months prior to the spring bloom, indicating that sea ice melt-related blooming is not the mechanism at work here.

Note that Chl-a concentrations at the exact site of the mooring (i.e. the location of the ADCP backscatter measurements) might remain low while increased Chl-a concentrations are observed nearby (electronic supplementary material, figure S6). This implies that while potentially vociferous

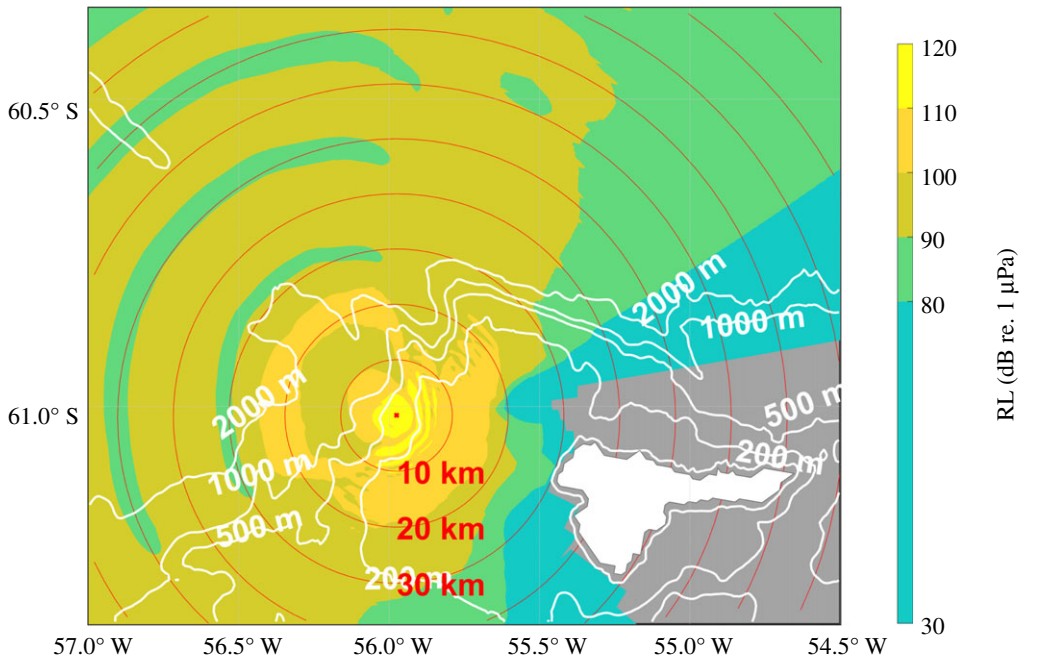

**Figure 7.** Sound pressure levels at a depth of 21 m as received from a virtual source placed at the recorder's position at 210 m depth (SL = 180 dB between 18 and 22 Hz) using sound speed profile #30 (winter) of Argo float WMO 7900409. This set-up serves as a model proxy for the real, reverse situation, i.e. a source at 21 m and the recorder at 210 m depth. Map created using Arndt *et al.* [30,31] and Greene *et al.* [32].

whales might feed on krill or fish associated with these nearby patches, prey species associated with the Chl-a patches might not show up in the ADCP backscatter data.

### 3.2.4. Sound propagation

Sound propagation modelling (employing a wintertime sound speed profile) reveals a highly anisotropic detection range for the AWI-EI recorder location. Figure 7 colour codes the location of hypothetical 180 dB re 1 µPa$^2$ m$^2$ sources (virtual fin whales) by their respective RL as they would be recorded at the AWI-EI recorder. A clear preference for signals originating from the semicircle spanning southwestwardly to north northeastwardly directions is observed, whereas the shallow topography southeastward of the AWI-EI recorder significantly impairs receiving signals from that region.

Sound with RLs higher than 110 dB re 1 µPa originate from closer than 10 km (yellow), those with 110 dB re 1 µPa > RLs >100 dB re 1 µPa from within 25 km around the recorder (orange); 90 dB RLs might originate from up to 350 km, however, for southwestern to northwestern directions only.

Results do not differ qualitatively or quantitatively when replacing the wintertime sound speed profile by a summertime profile (which features the shallow sound speed minimum around 50 m depth typical for summertime); see electronic supplementary material, figure S7.

## 4. Discussion

The above results reveal a pronounced, repetitive seasonal pattern in fin whale acoustic occurrence in the EI region. The discussion hereinafter (i) identifies the period of significant usage of the EI region by fin whales, (ii) examines why the acoustic occurrence starts significantly later than the physical presence, (iii) conjectures on their corresponding breeding grounds, (iv) investigates the possible effect of sea ice on their behaviour, and (v) tentatively attempts to extract information on relative fin whale abundance trends from FINs. To this end, we visually compare the time series of fin whale acoustic occurrence and environmental variables presented above, as well as consider supplemental information on visual sightings and deductions given in the literature.

**Table 6.** Timing (columns 'acoustic presence' and 'acoustic peaks') and intensity (columns 'max. calls detected', 'CAB' and 'daily averaged FIN') of fin whale 20-Hz pulses. In 'acoustic presence' months without recording effort are indicated by dots, whereas recorded month without acoustic presence are marked by dashes. Some site names are given in figure 1. References: [25–27,50]. Analysis types: ASC: automatic spectrogram correlation; FIN21, FIN89, FIN86 Fin whale index for different in-band versus off-band signal-to-noise ratios; MSC: manual screening. CAB: call abundance per hour calculated. Dashes indicate unavailable data or not applicable metrics.

| site | study | acoustic presence | analysis type | acoustic peaks | max. calls detected | CAB [h⁻¹] | daily averaged FIN |
|---|---|---|---|---|---|---|---|
| S2 | SI 2004 | -FMAM------- | ASC | 1 May and 2 May | 6000 and 6000/8 days | 31 and 31 | f = 1.8 and 1.8 → 3 dB |
|  |  |  | FIN89 | — | — | — |  |
| S1 | SI 2004 | -FMAM------- | ASC | 1 Apr and 2 May | 3500 and 3500/8 days | 18 and 18 | f = 3 and 3 → 5 dB |
|  |  |  | FIN89 | — | — | — |  |
| S1/WAP | SI 2009 | ..MAMJ------ | ASC | 3 Jun | 300/8 days | 2 | f = 1.6 → 2 dB |
|  |  |  | FIN89 | 3 Apr | — | — |  |
| **EI-AWI** | **this study** | **--MAMJJ-----** | **MSC** | **—** | **4301/1000 min** | **258** | **15, 14 and 14 dB** |
|  |  |  | **FIN21** | **13 May, 14 May and 15 May** | **—** | **—** | **14, 12 and 11 dB** |
|  |  |  | **FIN86** | **13 May, 14 May and 15 May** |  |  |  |
| EI-SCRIPPS | BP 2015 | ..MAMJJ...... | FIN21 | — | — | — | 22 dB |
|  |  |  | — | 14 Mar–14 Jul | — | — |  |
| Scotia Sea | SI 2009 | -FMA........ | ASC | 3 Mar | 12 000/8 days | 63 | — |
|  |  |  | FIN89 | 3 Apr | — | — | f = 1,2 → 1 dB |
| sightings near EI | diverse | JFMA--------D | — | — | — | — | — |
| Chile HA03 | SB 2019 | ---AMJJASON- | ASC | — | 30, 30 and 50 | 30, 30 and 50 | — |
|  |  |  | FIN-17 | 7 Jun and 8 Jun and 9 May | — | — | 5, 5 and 10 dB |
|  |  |  | FIN-85 | — | — | — | 2, 2 and 3 dB |
| Chile HA03 | SB 2019 | ---AMJJASON- | ASC | — | 70, 100 and 60 | 70, 100 and 60 | 10, 12 and 8 dB |
|  |  |  | FIN-17 | 14 Aug, 15 Sep and 16 Aug | — | — | 5, 6 and 5 dB |
|  |  |  | FIN-85 | — | — | — |  |

## 4.1. When do fin whales use the Elephant Island area?

Our acoustic data show an annually recurring, steady increase of acoustic power in the fin whale band starting in mid-February (late austral summer), reaching its maximum in May (austral winter), to be followed by a steep decline in June/August after which the acoustic power vanishes in the fin whale band until February of the following year. For this latter period, fin whale acoustic occurrence (DAO-AU) is observed only sporadically, yet calls continue to occur intermittently. Our observations of fin whale acoustic occurrence are consistent with earlier acoustic studies conducted near EI and farther west, offshore the Antarctic Peninsula, suggesting that these sites form a possibly contiguous area of abiding ecological relevance to fin whales, from December (anticipating the knowledge gained from visual sightings referred to below) to August, at least.

Baumann-Pickering et al. [29] collected recordings at EI-SCRIPPS (figure 1) between March and July 2014 in close proximity to our mooring. Missing the February onset, their FIN21 (table 4) starts strong in March 2014 and remains at around +16 dB until May before decreasing throughout June and July. Farther afield (west of the Antarctic Peninsula) and about a decade earlier Širović et al. [27,28] showed a strong seasonal occurrence of fin whale calls from March to June, peaking in April–May at locations S1/WAP and S2 (table 6). Additionally, recordings from a more easterly site in the Scotia Sea operated between January and April 2003 [28] showed fin whale acoustic occurrence from February to April (Scotia Sea, table 6).

Visual sightings have been reported for the period December–April (electronic supplementary material, figure S8), which, however, might be biased, i.e. constrained both in its beginning and its end, by the times when tourist ships visit these waters, i.e. austral summer. Scientific surveys during these periods [10,13,80–82] indicate fin whales being involved in feeding activities and occurring in large groups. Visual and acoustic detections combined hence suggest EI area to be highly frequented by fin whales from December to August.

## 4.2. What drives their acoustic display in the Elephant Island area?

With a more or less concurrent onset of fin whale acoustic power at all sites along the Antarctic Peninsula occurring months after their physical arrival, the question arises, what does actually drive this acoustic crescendo? The following discussion will show, that, rather than being feeding related, as one might expect to be the case on a major feeding ground, this call's increase is likely to be linked to the commencement of the breeding period.

One cue supporting this interpretation derives from the all but entire absence of 20 Hz pulses prior to March, although visual surveys and whaling data reported fin whales to be abundantly present in the region during this period [13,80–82]. Santora et al. [10] reported foraging fin whales in January for several years, clustering in areas were krill was simultaneously present. Peaks in late January/early February in our acoustic backscatter data also point towards these months being a likely feeding period (table 5 and figure 3), as do the phytoplankton blooms observed during the January–March period (cf. electronic supplementary material, figure S6). The fact that effectively no 20-Hz pulses were recorded in this period, in spite of ample biomass availability and whales being sighted within the listening range of the recorder (albeit, in some cases, in different years than the acoustic recordings), indicates that the whales do not produce this call type during feeding.

Seeking further clues regarding the role of the 20-Hz pulse, we investigated its diel pattern. Diel calling patterns in vocal behaviour have been documented for baleen whale species [83,84], including various fin whale populations worldwide, with the daily timing of peaks in calling activity varying between oceans according to feeding patterns and prey species [47,51,85]. Evidence of a diel pattern in fin whale acoustic activity at EI could hence be viewed as indicative of 20-Hz pulses being feeding related. Our acoustic data, however, show no such pattern throughout the entire period of fin whale acoustic occurrence (March–August), regardless of whether the daily migration amplitude (see electronic supplementary material, figure S5) indicates a lacking (April/May) or discernible (May–July) diel vertical migration of potential prey, supporting the conjecture that the production of 20-Hz pulses is not directly related to feeding.

Contrastingly, the May peak in call activity off EI falls well into place with the timing of the mating season of Southern Hemisphere fin whales, suggesting that the 20-Hz pulse might rather be produced in a mating context. While we lack direct evidence of such a connection, 86% of conceptions in fin whales are reported to occur between April and October [86–88], a time-span largely overlapping with the primary call period March–August. April/May has furthermore been reported as the season of

increased spermatozoa production in male fin whales caught off South Georgia [86]. For the northern hemisphere, a contemporaneous match of the estimated conception time has also been reported for singing fin whales on Arctic feeding grounds during boreal autumn and early winter [51].

Worth noting in this context is the observation that humpback whales show the peak of vocalization intensity being related to hormonal changes [89,90]. In analogy, increased vocal activity by male fin whales could be attributable to hormonal state (e.g. elevated testosterone levels, [43]), rather than to momentary prey availability. Instead of following a diurnal feeding and singing rhythm, fin whale males off EI might alternate feeding and singing behaviour opportunistically throughout the day in response to their particular nutritional and hormonal state [91–93].

## 4.3. Where do they go, where do they come from?

Multi-year data from Juan Fernandez off the Chilean coast showed a robust seasonal pattern in acoustic occurrence with most fin whale call detections between June and September, with a clear peak in August (table 6, [50]). The notion of fin whales recorded at both locations belonging to the same population is supported by the close match of the HF-component of the recorded calls. Gedamke & Robinson [45] suggested that the HF-component co-occurring with the 20-Hz pulse is a characteristic trait of specific fin whale populations. With the HF-component recorded by us between 2013 and 2015 at EI being at $85.6 \pm 1.5$ Hz, these fin whales may hence well belong to the same population as those from the Chilean waters, featuring an HF-component of $85.3 \pm 0.8$ Hz [50]. This finding mirrors the notion of a fin whale population migrating between central Chile and the western part of the Atlantic sector of the Southern Ocean on the basis of fin whales taken there, having previously been tagged off central Chile during whaling times [94].

The ongoing acoustic occurrence beyond the start of the estimated conception time suggests that not all fin whales may immediately migrate north to lower latitude waters to mate, but rather remain at high latitudes or migrate later, possibly to exploit food niches that are not available at lower latitudes while mating. Lingering late on feeding grounds might actually be an advantageous strategy for males, as they could in this way also gain access to non-migratory females that remain in the feeding area or that migrate later, as has also been hypothesized for humpback whales [89]. Historical catch data suggest that migration is not obligatory for fin whales [95] and that the timing of migration may differ between sex and age classes [87]. Mizroch *et al.* [96] and Geijer *et al.* [97] presented evidence that across different stocks and areas, fin whales are likely to exhibit much more complex and variable movement patterns, defying the traditional migration model that breeding occurs exclusively in lower latitude wintering grounds (e.g. [80]).

## 4.4. Does sea ice affect physical presence in the Elephant Island area?

Sea ice, by contrast, appears to be, at least on the basis of our data, a rather unlikely driver of fin whale acoustic occurrence and their migratory behaviour. In our data, fin whale acoustic occurrence subsides clearly before the onset of any sea ice formation (figure 3d), indicating that the decrescendo appears not to be caused by a growth in sea ice 'pushing' the animals out of the area. This lagged ice growth relative to the cessation of fin whale calling is also clearly evident in Širović *et al.* [27], their figure 6, panel S1 and S2 in particular. While the authors note a (statistically insignificant) negative correlation between fin whale calls and sea ice, their actual time series exhibit, like ours, a distinct lag between the decrease in acoustic activity and the sea ice growth, suggesting the former occurring independently of the latter.

Mackintosh [81] assumed that northward migration might be triggered by the depletion of krill stocks or sufficient food consumption for the season rather than the advancing of the pack-ice. The commencement of the breeding season might form a further factor provoking migration independent of the sea ice situation. For Arctic waters, Simon *et al.* [51] pointed towards secondary oceanographic effects on prey availability rather than the physical presence of sea ice as a limiting factor in fin whale distribution. With regard to the southward migration, given that the 30 km radius around our recording site was never fully covered by sea ice, it appears unlikely that their arrival on the feeding grounds is impeded by sea ice.

## 4.5. Relating abundance trends to trends in acoustic occurrence

A most pressing question is whether fin whale populations have recovered since the implementation of the IWC's whaling moratorium and if acoustic data can contribute to monitoring changes in abundance. While

capturing abundance (trends) of pelagic whales with visual surveys appears prohibitively costly, PAM might reveal long-term trends or at least relative acoustic abundance more realistically. Here, we seize the opportunity of having acoustic data from recorders deployed concurrently nearby and consecutively farther astray over the period of about one decade, to explore such trends and their validity.

The CAB = 258 h$^{-1}$ observed in May 2013 is the highest reported in any of the fin whale studies considered in this paper. The result is mirrored by the FIN21 and FIN89 indices peaking at around 13 dB, which ranks as second highest reported FIN index, while Baumann-Pickering et al. [29] present a peak FIN21 in March of 22 dB (the study does not analyse for CAB, preventing a direct comparison thereof) for the nearby EI-SCRIPPS recorder. Farther afield, Širović et al. [27,28] present CABs reduced by about an order of magnitude and FINs by about 10 dB less for both sites S1/WAP, S2 and SS at distances of 192 and 353 NM to the west and 134.9 NM to the east from our site (table 6). Worth noting, in this context, is the fact that FIN indices, i.e. effectively neighbouring on-band to off-band SNRs are robust against flaws in the acoustic recorders' calibration, as this affects both bands the same way. Hence a numerical comparison of FINs from different recordings appears possible, even without high-precision recorder calibrations.

Extracting the reasons for the order of magnitude increase in call activity over the course of 10 years remains challenging at this time, as both a growing of the population [14,19], but also a re-occupancy of specifically EI by a population (or any mix thereof) might be the underlying cause. Unfortunately, information on current abundance trends in Southern Hemisphere fin whale population is presently unavailable (see [24] for further details), making it difficult to resolve between the two. The latter might reflect on younger animals' regaining of a cultural memory regarding a preferential usage of this area, a knowledge that might have survived the whaling times in only a few specimen. Alternatively, a re-occupancy of the EI area might be linked to '…alterations in prey availability…' or '…increasing … whale abundance intensifying pressure on prey availability elsewhere…', as suggested for humpback whales by Findlay et al. [98]. These assertions appearing quite plausible, noting the response of fin whale prey species to climate change ([99] for a review, [100,101]), which is particularly strong along the Antarctic Peninsula, and predictions of negative future impacts of climate change on all Southern Ocean whale species, especially for Pacific and Atlantic/Indian fin whale populations by Tulloch et al. [102].

It cannot be excluded that specific analytical approaches result in these stark differences between the two Širović et al. studies [27,28] on the one hand, and the Baumann-Pickering et al. [29] and our study on the other hand. Širović et al. present their data as daily averaged in-band to out-band ratios of spectral density, while Baumann-Pickering et al. [29] and we present daily averaged in-band and out-band differences of spectral density levels. Formally these should be related by

$$\text{spectral density level difference} = 10 \times \log_{10}\left(\frac{\text{in-band spectral density}}{\text{out-band spectral density}}\right),$$

providing the results given in table 6, last column. With CAB differing by about an order of magnitude, which results in a FIN difference of $10 \times \log_{10}(10) = 10$ dB, CAB data and FIN data, while independently derived, are consistent within each group of studies, giving reason to trust the findings and seek an explanation beyond the mere analytical approach.

Nevertheless, additional factors affecting the background noise level, like recorder distance to sea floor and sea surface as well as local topography, as well as sound propagation will also affect FIN and CAB, prohibiting, at least for now, any irrevocable conclusions. These limitations are discussed below in §4.6.3 'Implications of recorder placement'.

## 4.6. Confounding factors

The validity of the scenarios developed above as a likely explanation for the observed seasonal patterns in acoustic activity is subject to possible confounding factors that may result in similar acoustic trends. The following discussion addresses these methodologic complexities with regard to their potential impact on the metrics used herein, FIN, CAB and DAO.

### 4.6.1. Implications of sound propagation

Changes in temperature or sea ice cover affect the acoustic propagation of signals underwater [46]. In our case, seasonal changes in acoustic propagation can, with relative certainty, be excluded as a driving factor for the observed seasonal increases in the FIN21-AU, FIN86-AU and LTS-AU. Changes in sound speed profile (primarily the formation of a surface duct during the summer months) were found to only

insignificantly impact the detection range (electronic supplementary material, figure S7). In fact, the formation of a surface duct would affect the acoustic occurrence opposite to what is observed: a surface duct leads to slightly increased detection radii in January–March, while the acoustic occurrence (both SNR and CAB) was near its minimum during this period. Additionally, CAB also confirmed that the increase in SNR was explicable by an actual increase in call abundances and not call amplitude (which would have been indicative of possible propagation effects).

The presence of sea ice, on the other hand, is known to reduce sound propagating ranges. However, even at the time of onset of sea ice presence (early August), fin whale acoustic occurrence has already largely vanished, rendering a discussion of the acoustic impact of sea ice irrelevant.

### 4.6.2. Implications of call type selection

Deriving physical presence from acoustic data introduces the risk of confusing acoustic absence with physical absence, producing, in technical terms, false negatives. Passive acoustic data collected in this study clearly lags the peak reported in visual sightings (e.g. [13]), similar to observations from southern California and northern Chile, 'where the peak in 20-Hz song presence also lags visual sightings' ([47] and S. Buchan 2020, personal communication). This confirms once more that call abundance or acoustic energy cannot be equated to physical presence, which is why both visual and acoustic data are consulted in this study to inform on fin whale physical presence.

One may wonder if additional insight could have been gained by including further fin whale calls. In this study, the four measures of fin whale acoustic occurrence rely solely on the occurrence of the 20-Hz pulses and its 86 Hz HF-component, as these are considered the most reliable (in the sense of minimal false positives) indicators for fin whale presence. While 20-Hz pulses are apt for CAB estimates, since it is easy to distinguish these from blue whale Z calls due to the latter's unique shape, the 86 Hz HF-component, or better its energy, clearly stands out in calculating FIN against the ambient sound in the adjacent frequency bands. Please note, that the 20-Hz pulse proper is less well suited for FIN calculations due to the energetic contributions from the Z-call, while the FIN86 allows direct comparison of e.g. call rates even between studies (e.g. [75,103]), as discussed in the previous section on abundance trends.

Alongside the 20-Hz pulses (with or without HF-component), two other call types that have been (tentatively) assigned to fin whales occurred also occasionally in our data: a low-frequency call around 13 Hz [104] and 40 Hz feeding calls [36]. These could have provided additional information on fin whale presence when either no 20-Hz pulses are produced, or during lack of sightings. However, the 13 Hz calls occurred rarely in the SonoVault data ($n = 513$ versus 25 568 for the 20-Hz pulses between March and June 2013) and have not been attributed to fin whales with certainty. The 40-Hz feeding calls were excluded from further analysis as these are known to be produced by several baleen whale species [105–107] and are difficult to attribute to a species with certainty unless scrutinized in detail. These calls were therefore likely to produce little new, potentially false positive, presence information, which is why we did not consider these in the analysis. Additionally, mooring noise in phase with tidal patterns regularly contaminated this frequency range. Should our disregard of the 13 Hz calls and 40-Hz calls result in us falsely considering the months September–November of being sparsely populated by fin whales at EI, it would only imply that our conclusion on the importance of EI in fin whale ecology would gain even more weight than already described in this study.

Finally, a particularity of the 20-Hz pulse requires mentioning. The 20 Hz pulses are produced in two different contexts. Regular 20-Hz pulse sequences (song) are thought to be related to breeding contexts, serving mate attraction or advertisement of resources [39,43]. Irregular calls and counter-calls are believed to serve a function in general social behaviour, maintaining contact between individuals [38]. It is conceivable that one of these features a diel cycle while the other does not, which may mask the diel cycle of the former. While the analyst of the data tended to classify most of the 20-Hz pulses as being part of song, the high density of 20-Hz pulses in our data made a clear distinction impossible. While this does not affect the meaning of the metrics DAO-AU, LTS-AU, FIN21-AU, FIN86-AU and CAB-SV, it cannot be excluded that a diel pattern may exist in song or irregular counter-call sequences (e.g. [108]), which became obscured by merging both calls into one metric.

### 4.6.3. Implications of recorder placement

Recorder location and mooring noise cannot be excluded as a factor affecting detection range and hence CAB and FIN. With all Antarctic recorders deployed along the shelf slope, topography (figure 1) takes an

essential role in shaping which parts of the ocean are eavesdropped on. Recordings at starkly varying recorder depths (between 260 m and around 3000 m) are affected by differences in SNR and sound propagation (electronic supplementary material, figure S7), though one would have expected that SNRs (i.e. FINs) are lower for recorders nearer the surface and coastline, due to their proximity to breaking waves and hence background noise. Alternatively, Širović et al. [27] mention '*poor performance of the automatic call detection method during periods of intense calling*' and suspect that it rather underestimated the total call number (see their p. 2332).

The slight differences in acoustic intensity for the various recorders along the Antarctic Peninsula and Island chain might be attributed to interannual variability, small-scale migratory patterns, or to differences in their geographic settings. However, the difference in onset between our recorder and the EI-SCRIPPS ([29] table 6), which overlap in time cannot be attributed to interannual variability. While our recorder recorded a clear crescendo of both FIN21 and FIN89 from null to about half-maximum intensity, the recorder at EI-SCRIPPS captured near maximum levels throughout March. Most likely, the recorders 'observed' different regions due to local topographic effects. Our recorder was located on the shelf at 210 m depth (water depth 320 m), detecting whales primarily located northwest of it (electronic supplementary material, figure S7), while the recorder deployed by Baumann-Pickering et al. [29] resides on the shelf break at about 760 m water depth. Reflecting on these different geometries, propagation modelling resulted in a much shorter detection range for this deeper recorder. However, propagation modelling does not consider the impact of background noise levels varying between a shallow deployment with impact from coastal surf, and a deeper deployment farther offshore. Resolving this issue probably requires the deployment of a dedicated mooring at the EI-SCRIPPS site, with recorders at both depths. In conclusion, to be able to obtain FINs comparable across years and decades, it is necessary to maintain recording sites and set-ups, to ensure reproducible sound propagation and noise conditions.

# 5. Conclusion and outlook

Our study shows that the EI region provides an important habitat for Southern Hemisphere fin whales throughout a substantial part of the year (at least December to August). Whether the high acoustic activity off EI merely serves as a prelude to later mating or if such actually occurs already around EI remains obscure for now, yet it is clear that EI represents an important region to fin whale sustenance, warranting special protection of its ecosystem. This role of EI should receive heightened attention, also in the light of it being a major krill fishing and touristic area. Management of these activities appears to be necessary to ensure the fin whale's further unhindered recovery. Moving in this direction, the IUCN Marine Mammal Protected Areas (MMPAs) Task Force recently recognized EI as a candidate important marine mammal area (IMMA) and the International Whaling Commission's Southern Ocean Research Partnership (IWC SORP) facilitated further research on Southern Hemisphere fin whales in the area to determine their status. These efforts are particularly important in light of a changing climate which additionally influences baleen whale recovery especially in rapidly warming regions of the Southern Ocean.

One breeding ground for these whales appears to be located in the Juan Fernandez Archipelago (JFA) off the Chilean coast as suggested by the similar HF-components recorded at both locations and by the plausible lag in the progression of acoustic activity between the two sites. Future research could corroborate this finding by concurrently deploying PAM recorders along the presumed migratory corridor; however, direct evidence of a connection between EI, JFA and potential further northern breeding grounds would best be established by tracking whales during their migration. Optimally, tags would, in addition to position, monitor for vocalization of the tagged animal in the 50–150 Hz band only, minimizing demands for data transmission and the tag's embedded acoustic recorder sensitivity. Last but not least, biopsy and eDNA sampling of fin whales at either location could be a desirable asset to gain information on population affinity. However, acoustic research should also look beyond EI to corroborate the hypothesis that the HF-component is population specific, which, if established, would provide a powerful method to monitor the development of fin whale populations.

A marked increase in both FIN and visual sightings near EI across the first two decades of this century is evident from this and other studies, providing hope that fin whales are finally growing in abundance. Fin whales in particular are still recovering from severe exploitation during last century's whaling activities, hence understanding their abundance trajectory is important to research and management decisions of the IWC. Comparing FIN between subsequent studies might offer a way to infer such

information on abundance trends, at least qualitatively; however, the method requires further development and standardization. Apart from observational caveats, like the recorder placement affecting the recorder's listening radius and ambient noise, which both affect FIN directly, changes in local FIN may be caused by multiple ecological changes, like abundance, usage of the study region, or even an altered acoustic behaviour, which are hard to disentangle. Probably a combination of research methods at hand, i.e. visual sightings, acoustic monitoring, tagging and DNA analyses will be needed to obtain robust estimates of fin whale abundances, their trends and the underlying drivers.

Ethics. Permission to deploy moorings in the Scotia Sea was granted by the German Federal Environmental Agency (UBA) under permits I 3.5-94003/286 and II 2.8-94003-3/347.

Data accessibility. The sound analyses data and code are available through the Dryad Digital Repository (https://doi.org/10.5061/dryad.cjsxksn3m) [109]. Sea ice concentration data through the University of Bremen (http://www.iup.uni-bremen.de/seaice/amsr/). Chlorophyll data are accessible online through Ocean Colour Climate Change Initiative dataset, v. [3.1], European Space Agency, at https://esa-oceancolour-cci.org/. Data for sunrise, sunset and nautical twilight are available from the US Naval Observatory sunrise/sunset and nautical twilight table https://www.usno.navy.mil/USNO/astronomical-applications/data-services.

Authors' contributions. E.B. coordinated the study, carried out part of the data analysis and drafted and wrote the manuscript. O.B. carried out the acoustic data analysis and helped drafting the manuscript. I.V.O. contributed to the drafting and writing of the manuscript. K.T. provided the sound propagation analyses and contributed to the internal scientific discussions. B.C. carried out data analysis on ADCP and chlorophyll and wrote the instrument-specific sections. R.M., M.M., E.S. and S.Z. carried out data analysis and contributed to the discussion around the manuscript. S.S. deployed the recorders, cleaned up the raw data and contributed to the discussions around the manuscript. All authors gave final approval for publication.

Competing interests. The authors have no competing interests.

Funding. The authors carried out the fieldwork and data analysis while employed at the Alfred Wegener Institute Helmholtz Centre for Polar and Marine Research, Bremerhaven, Germany. B.C. contributed to the paper while employed at Thünen-Institute for Sea Fisheries, Bremerhaven, Germany.

Acknowledgements. We thank crew and officers of RV Polarstern expedition ANT XXIX/2 together with Matthias Monsees, Rainer Graupner, Wolfgang Zahn and Gerd Rohardt for the deployment of the mooring, as much as crew and officers of RV Polarstern expedition PS96 and Michael Schröder and Andreas Wisotzki for the recovery of the mooring. We thank two anonymous reviewers for their detailed, insightful and constructive comments.

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
