## [Peer Review File · Royal Society Open Science]

Review History

RSOS-201142.R0 (Original submission)

Review form: Reviewer 1

Is the manuscript scientifically sound in its present form?

Yes

Are the interpretations and conclusions justified by the results?

Yes

Is the language acceptable?

Yes

Do you have any ethical concerns with this paper?

No

Have you any concerns about statistical analyses in this paper?

No

Recommendation?

Major revision is needed (please make suggestions in comments)

Comments to the Author(s)

This is a really comprehensive manuscript presenting data from an area that could greatly benefit from this increased insight and information. The component parts of the manuscript are well written but the ordering is very confusing and needs to be clarified. It is hard to follow the sequences of analyses and why, and at times the text could be reduced so as to synthesize the information better. Both the discussion and conclusions should be shortened. Additionally, the methods need reordering and reorganizing. However, once these things are done and the manuscript is more accessible to the reader, it will definitely be a really valuable publication.

Review form: Reviewer 2**Is the manuscript scientifically sound in its present form?**

Yes

Are the interpretations and conclusions justified by the results?

Yes

Is the language acceptable?

Yes

Do you have any ethical concerns with this paper?

No

Have you any concerns about statistical analyses in this paper?

No

Recommendation?

Major revision is needed (please make suggestions in comments)

Comments to the Author(s)

Please see document attached (Appendix A).

Decision letter (RSOS-201142.R0)

Dear Dr Burkhardt

The Editors assigned to your paper RSOS-201142 "SEASONAL AND DIEL CYCLES OF FIN WHALE ACOUSTIC PRESENCE NEAR ELEPHANT ISLAND, ANTARCTICA" have now received comments from reviewers and would like you to revise the paper in accordance with the reviewer comments and any comments from the Editors. Please note this decision does not guarantee eventual acceptance.

Please submit your revised manuscript and required files (see below) no later than 21 days from today's (ie 13-Oct-2020) date. Note: the ScholarOne system will 'lock' if submission of the revision is attempted 21 or more days after the deadline. If you do not think you will be able to meet this deadline please contact the editorial office immediately.

on behalf of Dr Asha de Vos (Associate Editor) and Pete Smith (Subject Editor)
openscience@royalsociety.org

Associate Editor Comments to Author (Dr Asha de Vos):
Comments to the Author:

Thank you for your submission. The reviewers have done a stellar job at providing detailed input to help make this manuscript better. In its present state, it appears to lack key components like a specific scientific goal and an overall study approach that then perhaps hinders the focus of your writing. Please consider all the comments provided as they will certainly improve your manuscript and make it publishable. Thanks

Reviewer comments to Author:
Reviewer: 1

Comments to the Author(s)

This is a really comprehensive manuscript presenting data from an area that could greatly benefit from this increased insight and information. The component parts of the manuscript are well written but the ordering is very confusing and needs to be clarified. It is hard to follow the sequences of analyses and why, and at times the text could be reduced so as to synthesize the

information better. Both the discussion and conclusions should be shortened. Additionally, the methods need reordering and reorganizing. However, once these things are done and the manuscript is more accessible to the reader, it will definitely be a really valuable publication.

Reviewer: 2

Comments to the Author(s)

Please see document attached.

===PREPARING YOUR MANUSCRIPT===

===PREPARING YOUR REVISION IN SCHOLARONE===

Please ensure that you include a summary of your paper at Step 2 'Type, Title, & Abstract'. This should be no more than 100 words to explain to a non-scientific audience the key findings of your

research. This will be included in a weekly highlights email circulated by the Royal Society press office to national UK, international, and scientific news outlets to promote your work.

Author's Response to Decision Letter for (RSOS-201142.R0)

See Appendix B.

RSOS-201142.R1 (Revision)

Review form: Reviewer 1

Is the manuscript scientifically sound in its present form?

Yes

Are the interpretations and conclusions justified by the results?

Yes

Is the language acceptable?

Yes

Do you have any ethical concerns with this paper?

No

Have you any concerns about statistical analyses in this paper?

No

Recommendation?

Accept as is

Comments to the Author(s)

This is a nice a valuable paper, well done.

Decision letter (RSOS-201142.R1)

Dear Dr Burkhardt,

It is a pleasure to accept your manuscript entitled "SEASONAL AND DIEL CYCLES OF FIN WHALE ACOUSTIC OCCURRENCE NEAR ELEPHANT ISLAND, ANTARCTICA" in its current form for publication in Royal Society Open Science. The comments of the reviewer(s) who reviewed your manuscript are included at the foot of this letter.

Please see the Royal Society Publishing guidance on how you may share your accepted author manuscript at <https://royalsociety.org/journals/ethics-policies/media-embargo/>. After

publication, some additional ways to effectively promote your article can also be found here <https://royalsociety.org/blog/2020/07/promoting-your-latest-paper-and-tracking-your-results/>.

on behalf of Dr Asha de Vos (Associate Editor) and Pete Smith (Subject Editor)
openscience@royalsociety.org

Associate Editor Comments to Author (Dr Asha de Vos):
Associate Editor: 1
Comments to the Author:
Looking forward to the publication of this paper. Great work!

Reviewer comments to Author:
Reviewer: 1

Comments to the Author(s)
This is a nice a valuable paper, well done.

Appendix A

General Comments:

This manuscript presents very interesting data on fin whale acoustic presence and of particular interest, environmental correlates, off Elephant Island, Antarctica. This passive acoustic monitoring approach coupled with environmental data is of great value. This study presents important data over a three-year timeseries from a poorly accessible site in the southern ocean and thereby improves the coverage of both passive acoustic monitoring and oceanographic monitoring. I congratulate the authors on their effort and hard work for collecting this data and producing this study. It is not a small feat. My overall view is that this manuscript should be considered for publication after a major review.

Overall, the manuscript makes for difficult reading because the study goal and study approach are not thoroughly outlined, important information is supplied in footnotes and supplementary materials, and the discussion is too lengthy, and I think should be tightened up and aligned with study goals and key points the authors want to make. With quite a lot in footnotes and supplementary material, maybe the authors are trying to pack too much in? There are also quite a few minor annoyances, such as some stylistic issues with writing, acronyms not being properly spelt out, deficient figure and table captions, figures and tables not in the proper order or not quoted in the text, and others I point out in the detailed comments.

As mentioned above, motivation needs to be better outlined. There are studies (Širović; Baumann-Pickering) that examine seasonality and relation to environmental correlates, so it would be helpful to see, in the last section of the introduction or start of the methods, a clear outline of what your specific scientific goal is here and how it builds on what is already known or unknown in this area.

The last section of the Introduction and the Methods are sometimes hard to follow because the overall study approach is not clearly outlined clearly anywhere. I suggest the authors either provide an overview of the study approach in the last part of the introduction (which I suggest in one of the detailed comments), or maybe even better, at the beginning of the methods sections. This avoids new elements “surprising” the reader as the methods, results and even discussion are read. There are also parts of the methods section that need to be more complete and I have given examples in my detailed comments (e.g. details on units used, call count methods, discarding of data due to background noise, use of summer and winter sound speed profiles, etc).

Regarding the Supplementary Material, to me the propagation work is important enough that the authors should consider putting it in to the main text somehow (Table form even?). Also, the order in which supplementary figures appear in the text is not in the proper order.

One of the most interesting parts to me is the environmental data, but too much of the discussion is dominated by pretty speculative discussion of calling behaviour and context of calls, which cannot be resolved with the methods employed in this study (and arguably cannot be resolved with any methods currently available). I think the manuscript would benefit from putting a stronger focus on a) the relationship between calls and environmental data and b) the issue of calibrating calls to compare with previous studies, which is only introduced in the discussion.

For example for point a), the ADCP data is skimmed over in the results section and diel vertical migration is apparently not examined, which would be interesting here because the study

investigates diel calling. Maybe some statistics or timeseries analyses can be applied to examine links between calls and environmental correlates?

For point b) this a central topic at the moment, as the authors mention in the text, and they would be making a strong contribution to the field if they were able to outline the steps they took in the methods to compare results to other previous studies (Table 4) and to come to the conclusion that “CAB and FIN indicated a 10-fold increase over the past 10 years”.

I think the conclusion should also be tightened up so it is not too much more of the discussion but sticks to key findings and implications of these, and future projections.

Detailed comments: (page numbers according to pdf of proof)

P5 lines 33: rewrite “the area is hosting”, poor style

P5 line 48: last sentence of paragraph is poor stylistically

P6 lines 21: if they have scarcely been reported, can you say when they have been reported? Is this just anecdotal?

P6 lines 27: it is unclear from the text whether the previous studies [24,25] were year-round or not. I believe ref 24 resolves seasonality; therefore you need to say here why you are also looking at seasonality. Do you think presence is not longer seasonal? And is now year-round? Clarify motivation of study.

P6 l40: “this method facilitates examining whether previous hot spot observations should be judged as singular events or representing a more regular usage” what do you mean by hot spot observations??

P6 l46: “PAM data were analysed with respect to interannual, seasonal and diel patterns with the aim to improve understanding the ecological role of Elephant Island for fin whales as well as the function of the calls they produce.” You never mention the environmental data here. You should say, very briefly, your approach for understanding the ecological role of fin whales (which seems an ambitious goal to me and possibly beyond the scope of the paper) and the approach to determine the function of calls (which also seems like an ambitious goal).

P6 lines 26-51 I think this whole paragraph could be rewritten more succinctly to lay out more clearly your knowledge gaps and motivation, and your general objective.

P6 lines 53- p7 lines 6: I think this is a bit redundant. You don't have to say this. At best, you can put a sentence in the corresponding results section to explain the order of presenting the environmental data.

P7 l31: please provide a few words on why the 76.8 kHz frequency is considered adequate for the size of prey organism you are examining.

07 l10: Should be: "A 14-day running average was applied to the ensemble data". But also: why 14 days? Please provide justification for this choice of timescale.

P7 l22: circle and radius are redundant

P8 l13: Unit of sea ice cover?

P8 l20: you indicate supplementary figure 3, but have not yet mentioned 1 or 2? I believe these should be numbered in order of appearance in the text.

P8 l29. You should explain more about how you got that RL, not just put it in as a footnote, since this is a predominantly PAM paper.

P8 l18: is there any sediment data for this area? Since you are using the softwares default setting, I assume not, but you should state this clearly.

P9 l10: "an type Argo float" ?

P9 l22: same as for RL, you should say more about you SL than just a foot note. You should provide some references to support the choice of SL, which do exist for fin whales.

P9 l24: why 21m instead of 20m since you then say that you assume the whale to be singing at 20m. Also, can you provide any references form the literature that support the assumption of 20m? Because you describe your mooring later, the 210m comes out of nowhere, so you should say that 210m is deployment depth.

P9 l41: Insolation? This is a weird term to use here.

P9 l 44 you have not yet explained what call rate is. Also, what is "call rate abundance"? Call rate is calls per unit time, but what is the abundance part? Also, if you want to test something, you have to use statistics or explain that you will do it graphically.

I think the methods need to have a section at the beginning that outline your general approach. This will help the reader understand the methods.

P10 l16: do we still think that these might be contact calls? You should also briefly mention downsweeps. Why do you not consider downsweeps?

P10 l21: are you in the 89 o 99Hz region?

P11 l8: I don't think it is relevant to say how you collected your mooring elements, aside from the fact that these may be sources of noise, and if think that this is a problem for your later analysis, then state it here clearly. Otherwise delete.

P11 l10: why repeat the ADCP information here?

P12 l52: "perusal" is an odd choice of word

Table 2: hard to read and text cut-offs are akward.

P13 l8: your clarification on abundance vs rate should not go as a footnote. Also, your footnote is confusing. To me, abundance should be the total number of calls detected, and call rate is that number over time. Please improve your explanation.

P13 l24: you start with LTSAs but the list of metrics in the previous section is in another order.

Figure 3: Include units in the captions and acronym complete names. I suggest you remove the red lines, they are distracting and I would let the reader see for themselves. I suggest you put DAO in the panel where you report the fin index and not with sea ice cover. Also, you should reorganize the panels and put LTSA next to the fin index, and your environmental data next to each other.

P15 l10: I still don't understand why you are looking at a 14-day running average.

Section 3.3.3: Call counts is not an accurate title for this section. Please define difference between call counts and call abundance.

P15 l37 – p16 l15: this section should be tightened up or eliminated.

3.3.3.2: how did you visualize spectrograms? There are no details on this here.

4.1.1 The ADCP results section seems a bit superficial to me. Do you see diel vertical migration? This would be interesting to know since you are also looking at diel calling, I assume because you are interesting in the potential relationship between calling and DMV of prey.

Table 3: where is Table 3 referenced in the text???

4.1.3. Rather speculative and bit sounds more like discussion than results.

P19 l20 what do you mean by anisotropic?

4.1.4 This entire section describes results in a supplementary figure which makes me think that the figure should not be supplementary. Also, "sun's yellow" vs "bright yellow"?

P19 l44: you never mention in the methods that you use sound speed profiles from summer and winter. You only mention January CTD measurements.....

4.2.3: what is FINxx? You also use FINs and FIN index in the text. Please make sure you fully define all your acronyms. I assume you are referring to fin index but please be consistent and spell everything out.

4.2.4 The first paragraph is really more discussion than results.

4.2.4.2: What was your sample size for diel analysis? Same as previous section?

Figure 5: Time unit is weeks? The caption, units and axis title are confusing. These are weekly average fin whale calls per 10-min files? Also, this caption is the first mention that files are discarded due to background noise.

P23 l19: why the list of percentages? Can you not plot this?

P23 l 32 here is the first mention of statistics! You should have mentioned this in methods!

P25 l58 "Regular 20-Hz call sequences are thought to be related to breeding contexts, serving mate attraction or advertisement of resources" is irrelevant in this section. Why are you discussing the function of calls here? And I think you have already mentioned this a few times in the text.

Here you mention the 40-Hz downsweeps for the first time (I think you should mention them in the intro as well). What other species do they get confused with? I think you should be able to distinguish them from blue whale Dcalls because they are shorter.

P26 l49: this footnote should be in the results and should be discussed here.

P27 l 12: ref 60 does not refer to northern Chile and neither reference compares seasonality in sightings and acoustic presence.

P27 l17: where do you consult visual sighting data in this study?

P27 l46: rewrite “yet nevertheless it is”.

P27 l51: what do you mean by “traditional” in this context?

P28 l30 “supports the suggestion that call production is related more exclusively to a specific behavioural context rather than to environmental parameters”. This is a curious statement. I would say that call production is ALWAYS related to (acoustic) behaviour. Maybe what you are trying to say is that call rates here are responding to changes in call production/acoustic behaviour rather than changes in animal presence driven by environmental correlates. Is this what you mean?

Table 4: There is quite a lot of information lacking from the caption and this makes for difficult reading of this table. What is the last column “daily averaged FIN”? You also have to say in the text why you think the other sites in the table are relevant here, especially the mid-latitude site (Chile HA03), given that the calls do look different (peak frequencies at 17Hz and 85Hz).

P29 l 60: why you think the mid-latitude site (Chile HA03) is relevant here given that the calls are at different frequencies (peak frequencies at 17Hz and 85Hz)? From the text it looks like you assume that they are the same acoustic population.

5.2.1 and 5.2.2: I strongly suggest you tighten up this text and put more emphasis on the interesting environmental data you have here, rather than too much discussion on context of calls and super groups. I also suggest you rethink the titles of these sections, so they better reflect the points that you are making. I also think that your discussion on the function of calls should be linked to the fact that you see no diel patterns, indicating that these calls are not responding to DVM of prey and therefore likely not linked to feeding.

P29 l38: I think this paragraph is a bit speculative and could be reduced or eliminated. Same for the discussion about super groups that follows. I would strongly suggest you reduce this.

5.2.4 and 5.2.5 I would suggest these sections be expanded which 5.2.1 and 5.2.2 reduced.

P 35 l43. “Summarizing, we note that while CAB and FIN indicated a 10-fold increase over the past 10 years, confounding factors prohibit assigning this unequivocally to a like increase of population.” Yes, you are right that using acoustics to look at population trends is a critical area of research right now. This whole section is really interesting but I think you need to take it out of the discussion and put it in the results and present clearly in the methods how you are calibrating CAB

and FIN and comparing results from all three previous studies with your own. I think this would be a very valuable addition to the current knowledge if you can do this.

Conclusions: this need to be more concise and really just focus on your main conclusions and brief implications rather than too much more discussion about Antarctic krill.

Appendix B

Dear Dr.de Vos, dear reviewers,

Please find enclosed a clean and a track-changes version of the revision of our manuscript titled

*“Seasonal and diel cycles of fin whale acoustic occurrence near Elephant Island, Antarctica”
by Elke Burkhardt, Ilse Van Opzeeland, Boris Cisewski, Ramona Mattmüller, Marlene Meister,
Elena Schall, Stefanie Spiesecke, Karolin Thomisch, Sarah Zwicker and Olaf Boebel,*

for publication in Royal Society Open Science.

In this document you find our responses as to how we have addressed the suggestions and comments raised by the reviewers. The reviewers’ comments are in orange and our response is in blue. Also we adapted the title of the manuscript slightly by replacing “Acoustic Presence” with “Acoustic Occurrence” to better match the manuscript’s content.

The revision includes the major changes that were requested by the reviewers, which implied a restructuring of the manuscript and providing additional information in the methods and results section. With Word’s Track Changes Option marking any shift of text as “deleted” + “inserted”, this resulted in a rather hard-to-follow track-changes version, even though text was only moved between, e.g., results and discussion.

Thank you for evaluating our manuscript.

Sincerely,

Elke Burkhardt

Associate Editor Comments to Author (Dr Asha de Vos):

RSOS-201142 "SEASONAL AND DIEL CYCLES OF FIN WHALE ACOUSTIC PRESENCE NEAR ELEPHANT ISLAND, ANTARCTICA"

Please submit your revised manuscript and required files (see below) no later than 21 days from today's (ie 13-Oct-2020) date. Note: the ScholarOne system will 'lock' if submission of the revision is attempted 21 or more days after the deadline. If you do not think you will be able to meet this deadline please contact the editorial office immediately.

on behalf of Dr Asha de Vos (Associate Editor) and Pete Smith (Subject Editor)

Comments to the Author:

In its present state, it appears to lack key components like a specific scientific goal and an overall study approach that then perhaps hinders the focus of your writing.

Please consider all the comments provided as they will certainly improve your manuscript and make it publishable. Thanks

Reviewer comments to Author:

Reviewer: 1

Comments to the Author(s)

This is a really comprehensive manuscript presenting data from an area that could greatly benefit from this increased insight and information. The component parts of the manuscript are well written but the ordering is very confusing and needs to be clarified. It is hard to follow the sequences of analyses and why, and at times the text could be reduced so as to synthesize the information better. Both the discussion and conclusions should be shortened. Additionally, the methods need reordering and reorganizing. However, once these things are done and the manuscript is more accessible to the reader, it will definitely be a really valuable publication.

Reviewer: 2**Comments to the Author(s)**

Please see document attached.

Comments Reviewer 1:

Abstract: This abstract doesn't really bring out the value of this project presently. It could really use some refocusing.

The abstract was reworked.

Introduction:

PG. 3 L 54 to Pg4 L6. This really doesn't work for me. The discussion can occur in whatever order you wish it to but it makes no sense at all for the most essential point, the PAM to come after the auxiliary information. It is confusing and does not read fluently.

We restructured the manuscript accordingly, i.e. present PAM related chapters prior to auxiliary information. The corresponding paragraph in the introduction was deleted.

Methods

Given the large number of different measurements made, it would help to have an introductory paragraph at the head of the methods stating broadly what was measured/collected and why e.g. PAM for long term with occasional other metrics including .. during ... season and ... during ... time period for the purpose of comparing In essence when were the sampling periods? Maybe bar graph could show over time when the different measurements were made versus PAM e.g.

We added an additional overview paragraph at the beginning of the methods section where we describe what kind of data was collected and for which purpose. We also added the suggested bar graph now Table 2 in the revised manuscript.

3.3. Data Analysis

The four different types of analyses that you mention here are in a given order. However you then mix up that order as you explain each of them. You need to stay organized and keep them in the same order.

We moved the LTS-AU up in the list and in Figure 3, which results in a consistent sequence between list, chapters, and figure.

Similarly these headers need to match the headers further down e.g. 3.3.3.

We promoted the subheadings 3.3.3.1 and 3.3.3.2 one level up to 3.3.3 and 3.3.4 so that the headings match those section 4.1, i.e. 4.1.3 and 4.1.4 as suggested.

Call counts does not match one of the 4 types of analyses mentioned at the top. I know it is the last one but it needs to match the wording, or else it gets confusing with these many types of approaches.

This became resolved with rearrangements described above. The section "call counts" actually contained information pertinent to both former sections 3.3.3.1 and 3.3.3.2, which is why we had opted for this subdivision. With the promotion of these subsections to 3.3.3 and 3.3.4, we now moved the general information given under "call counts" to the general information given at the beginning of the section 3.3 "Passive Acoustic Data Analysis", so "call counts" is not used in the revision,

Figure 5. This is difficult to read in terms of a seasonal plot. Maybe you can add an additional simplified plot that just indicated presence and has clear monthly axes, or you can improve the axes on this figure to make the seasonal time periods more obvious.

*Copied from response to reviewer 2, who provided a similar comment: Time tics advance by 6-days and represent the 24 h time period screened for calls (see section 3.3.4 (ex 3.3.3.1) for details). This format was chosen (and is maintained) as each tic represents an analysed day (i.e. **the effort**) even if the count was zero, like on the days in August-Sept-October. This “effort” information would get lost if only “monthly” tics were used. However, addressing the reviewers comment, we added a standard time axis on the top of the figure, along with color shading for the different seasons.*

Discussion

Your confounding factors should come at the end not the beginning. This section should start with your fin whale ecology discussion, after which you can remind readers of the caveats.

Reordered as suggested.

The discussion is long. I would suggest working to synthesize is a bit more so that it remains clear and to the point.

The discussion has been reordered and rewritten in large parts. It was shortened, yet also gained some length when including the requested details on the daily vertical migration and on potential biases when focussing the analyses on the 20-Hz pulse.

Comments Reviewer 2:

General Comments:

This manuscript presents very interesting data on fin whale acoustic presence and of particular interest, environmental correlates, off Elephant Island, Antarctica. This passive acoustic monitoring approach coupled with environmental data is of great value. This study presents important data over a three-year time series from a poorly accessible site in the southern ocean and thereby improves the coverage of both passive acoustic monitoring and oceanographic monitoring. I congratulate the authors on their effort and hard work for collecting this data and producing this study. It is not a small feat. My overall view is that this manuscript should be considered for publication after a major review.

Overall, the manuscript makes for difficult reading because the study goal and study approach are not thoroughly outlined,

Please see next paragraph for our detailed response to these general remarks. In general, we explicitly attempted to better outline the study's motivation, goal and approach.

important information is supplied in footnotes and supplementary materials,

We moved most (except for the very technical ones) footnotes into the text and the sound propagation plot to the manuscript proper, as suggested by the reviewer.

and the discussion is too lengthy, and I think should be tightened up and aligned with study goals and key points the authors want to make.

see response to general comments of reviewer 1 and response to detailed comment re sections 5.2.4. and 5.2.5 of reviewer 2 (towards the end of this response).

With quite a lot in footnotes and supplementary material, maybe the authors are trying to pack too much in?

We focused the manuscript more clearly by minimizing discussion on krill and super groups and removing all but very technical footnotes as suggested. However, as we seek to understand the ecological role of EI in fin whale live cycle, the issues of fin whale migration, acoustic behavior and stock evolution all require joint consideration and presentation of the underlying data in corresponding graphs.

There are also quite a few minor annoyances, such as some stylistic issues with writing,

We tried to resolve those mentioned by the reviewer.

acronyms not being properly spelt out,

The manuscript was checked for a complete description of acronyms.

deficient figure and table captions,

Figure and table captions have been checked and augmented where necessary.

figures and tables not in the proper order or not quoted in the text

Order of figures and tables as well as them being referred to in the text was checked.

, and others I point out in the detailed comments.

Attended to by working through the detailed comments.

As mentioned above, motivation needs to be better outlined. There are studies (Širović; Baumann-Pickering) that examine seasonality and relation to environmental correlates, so it would be helpful to see, in the last section of the introduction or start of the methods, a clear outline of what your specific scientific goal is here and how it builds on what is already known or unknown in this area.

We expanded the introduction by a new paragraph explicitly explaining the motivation of this study (understanding the role of Elephant Island in fin whale ecology) and outlined the differences in data acquisition (such as location and duration of the studies). To illustrate these differences early on, we moved the map of the study area up into the introduction and provide a new table (now Table 1) of the key deployment parameters of this and related studies. Please note that formatting did not allow to add this table in track changes mode.

The last section of the Introduction and the Methods are sometimes hard to follow because the overall study approach is not clearly outlined clearly anywhere.

We focused the introduction on the main motivation for this paper by providing a new last paragraph of the Introduction (see above) and by deleting the (previously) 2nd paragraph on large foraging groups (a topic that was deleted altogether) as well as the bulk of the (previous) last paragraph.

I suggest the authors either provide an overview of the study approach in the last part of the introduction (which I suggest in one of the detailed comments), or maybe even better, at the beginning of the methods sections to avoid new elements “surprising” the reader as the methods, results and even discussion are read.

We followed the reviewers 2nd suggestion and provide an overview of the study approach in the beginning of the methods section.

There are also parts of the methods section that need to be more complete and I have given examples in my detailed comments (e.g. details on units used, call count methods, discarding of data due to background noise, use of summer and winter sound speed profiles, etc).

See “detailed comments” section for specific response.

Regarding the Supplementary Material, to me the propagation work is important enough that the authors should consider putting it in to the main text somehow (Table form even?).

We followed the reviewer’s suggestion and added the figure for the winter sound speed profile to the main text as Figure 7, while retaining the summer situation in the Supplement. We also added the results for the EI-Scripps location to the Supplement (now Supplement Figure S7).

Also, the order in which supplementary figures appear in the text is not in the proper order.

Rearranged accordingly after revision and checked.

One of the most interesting parts to me is the environmental data, but too much of the discussion is dominated by pretty speculative discussion of calling behaviour and context of calls, which cannot

be resolved with the methods employed in this study (and arguably cannot be resolved with any methods currently available). I think the manuscript would benefit from putting a stronger focus on **a) the relationship between calls and environmental data**

Including time series of environmental data was motivated by our pre-study assumption that these likely affect fin whale physical and acoustic presence. To our surprise, however, peaks in MVBS, FIN and sea ice are clearly separated in time, hence a direct response of the fin whale's acoustic activity to prey availability or sea ice presence seems highly unlikely. In light of the clear separation of these peaks, applying more formal time series analysis tools to this data would be rather futile. However, we included information on (vertical) migration amplitude from the ADCP data, to further explore whether calls could be feeding related.

The observed temporal mismatch drives us to explore alternative driving factors behind the acoustic crescendo (breeding being the prime candidate) and the whales' departure for the northern breeding grounds the reason for the observed decrescendo. Given that a direct proof of this connections is hard to obtain – as the reviewer rightly states – the method of scientific reasoning appears to us as only possible alternative at this time. In doing so, we attempt to provide sound arguments while listing all conceivable confounding factors (which is a main reasons for the manuscript complexity and hence length), which is the basis of any good discussion.

and **b) the issue of calibrating calls to compare with previous studies, which is only introduced in the discussion.**

We moved the subclauses providing concrete values of FIN and CAB as derived by our study to the respective Results section. The methodology of calculating these values is described in the Material and Methods sections. We consider comparisons with data from other studies as well as evaluating the plausibility of changes in FIN versus changes in CAB to be well placed in the discussion, which is why we kept those parts there.

For example for point **a)**, the ADCP data is skimmed over in the results section and diel vertical migration is apparently not examined, which would be interesting here because the study investigates diel calling

We included additional information (text in the manuscript in chapter 3.4.1 and an additional figure in the Supplement Figure 5) on (vertical) migration amplitude from the ADCP data, to further explore whether calls could be feeding related.

Maybe some statistics or time series analyses can be applied to examine links between calls and environmental correlates?

See response above. The temporal mismatch between MVBS, FIN and sea ice concentration is absolutely evident, rendering any correlation analysis futile.

For point **b)** this a central topic at the moment, as the authors mention in the text, and they would be making a strong contribution to the field if they were able to outline the steps they took in the methods to compare results to other previous studies (Table 4) and to come to the conclusion that “CAB and FIN indicated a 10-fold increase over the past 10 years”.

The notion that CAB and FIN – at least as reported by the papers – have increased 10-fold is directly based on the values given in former publications and our manuscript, which are compiled in Table 7. A calibration between the studies was not performed, which is why we consider the information being complete. The discussion considers any confounding factors, which can only be overcome in future

studies by standardizing the mooring setups.

I think the conclusion should also be tightened up so it is not too much more of the discussion but sticks to key findings and implications of these, and future projections.

We reviewed the conclusion after finalizing the body of the manuscript and adjusted it accordingly,

Detailed comments: (page numbers according to pdf of proof)

P5 lines 33: rewrite “the area is hosting”, poor style

Changed to:” The area is known for high krill densities.....

P5 line 48: last sentence of paragraph is poor stylistically

This sentence is deleted.

P6 lines 21: if they have scarcely been reported, can you say when they have been reported? Is this just anecdotal?

Sentence replaces with more specific description including citations: “This increase in recent observations of high fin whale aggregations is particularly noteworthy. While large fin whale aggregations are mentioned in the secondary literature in general terms for highly productive areas [22, 23], original reports are lacking for the EI region during the post-whaling period prior to the turn of this century.

P6 lines 27: it is unclear from the text whether the previous studies [24,25] were year-round or not. I believe ref 24 resolves seasonality; therefore you need to say here why you are also looking at seasonality.

We added a new paragraph to the end of the Introduction which addresses these aspects and hopefully clarifies the motivation. Specifically, [24] resolves seasonality, but is located 192 nm to the west, representing a different habitat (shelf slope, rather than on-shelf). [BP, 25] on the other hand is located at EI but recorded from Mar-Jul 2014 only, and hence was not resolving seasonally or interannual recurrence.

Do you think presence is no longer seasonal? And is now year-round? Clarify motivation of study.

Our multi-year study concludes that presence is seasonal, yet quite long (Dec/Jan – Jul/August). The motivation for the study was not whether or not presence is seasonal (which was a fair a-priori assumption) but for how long exactly and how recurring fin whale presence on the EI shelf was/is.

P6 l40: “this method facilitates examining whether previous hot spot observations should be judged as singular events or representing a more regular usage” what do you mean by hot spot observations??

Deleted in process of rewriting this paragraph as requested by reviewers.

P6 l46: “PAM data were analysed with respect to interannual, seasonal and diel patterns with the aim to improve understanding the ecological role of Elephant Island for fin whales as well as the function of the calls they produce.” You never mention the environmental data here.

Quoted sentence replaced by a sentence explicitly mentioning the use of environmental data and its purpose: “To determine the temporal patterns of fin whale use of the EI shelf and to improve our understanding of this island’s role in fin whale ecology, we examined our passive acoustic data for interannual, seasonal and diel patterns and related these graphically to the seasonal patterns of environmental covariates as well as the timing of visual sightings”

You should say, very briefly, your approach for understanding the ecological role of fin whales (which seems an ambitious goal to me and possibly beyond the scope of the paper)

This seems to be a misunderstanding; the paper does not address the ecological role of fin whales (in the entire ecosystem) but the role of Elephant Island in fin whale ecology.

and the approach to determine the function of calls (which also seems like an ambitious goal).

The approach is now described in the sentence above (in the manuscript text), i.e. relating time series of fin whale acoustic presences to environmental time series and times of visual sightings.

P6 lines 26-51 I think this whole paragraph could be rewritten more succinctly to lay out more clearly your knowledge gaps and motivation, and your general objective.

The last paragraph of the Introduction was rewritten accordingly.

P6 lines 53- p7 lines 6: I think this is a bit redundant. You don’t have to say this. At best, you can put a sentence in the corresponding results section to explain the order of presenting the environmental data.

Deleted in Introduction, while Method section now commence with 2 summary paragraphs, graph and table, laying out the data sets used in this study.

P7 l31: please provide a few words on why the 76.8 kHz frequency is considered adequate for the size of prey organism you are examining.

We added “76.8 kHz, which corresponds to an acoustic wavelength of approx. 2 cm, i.e. the size class of euphausiids and small fish {Cisewski, 2016 #4418}. “ to the sentence.

P07 l10: Should be: “A 14-day running average was applied to the ensemble data”.

Changed accordingly

But also: why 14 days? Please provide justification for this choice of timescale.

Given that we seek to resolve seasonal patterns, 14 days appears a time-scale suitable of resolving such while at the same time smoothing of shorter-term fluctuations. For example, the Antarctic sea ice extent is well resolved at 14 day intervals (https://seaice.uni-bremen.de/data/amr2/today/extent_s_running_mean_amr2_previous.png).

P7 l22: circle and radius are redundant

We deleted “circle of”

P8 l13: Unit of sea ice cover?

Added “concentration” to title and unit [%] to first line of first paragraph

P8 I20: you indicate supplementary figure 3, but have not yet mentioned 1 or 2? I believe these should be numbered in order of appearance in the text.

Adjusted accordingly

P8 I29. You should explain more about how you got that RL, not just put it in as a footnote, since this is a predominantly PAM paper.

The footnote only addresses the fact, that the reference value used is $1\mu\text{Pa}^2$ (in line with the new ISO norm), rather than $1\mu\text{Pa}$, as some readers might expect due to previous (pre-ISO norm) practice (now deprecated by ISO 18405). The description of the calculation received levels has been added to section 3.2 Acoustic Data Acquisition:

“The AURAL recorded at 32768 Hz at a resolution of 16 bit [54]. The recorder was equipped with High Tech Inc. hydrophone HTI-68-MIN with a factory calibrated sensitivity of $S = -164 \text{ dB re } 1 \text{ V } \mu\text{Pa}^{-1}$. The Aural's system amplifier was set to $G = 22 \text{ dB}$ with the analogue digital converter's (ADC) peak voltage being 2V, resulting in a digital gain of $M = 84 \text{ dB}$. Data were stored lossless in 16 bit wav files and – based on the saved counts c , were converted to sound pressure levels according to

$$\text{SPL [dB re } 1\mu\text{Pa]} = 20 \cdot \log_{10}(c) - S - G - M = 20 \cdot \log_{10}(c) + 58$$

Calibration is based on factory calibration only, no further pre- or post-calibration was performed, nor did we apply any frequency-specific correction of hydrophone sensitivity. According to the manufacturer the recorder's frequency response is flat within $\pm 1 \text{ dB}$ over the usable frequency range.

P8 I18: is there any sediment data for this area? Since you are using the software's default setting, I assume not, but you should state this clearly.

We expanded this sections slightly by: “The propagation model solved for normal modes, employing a silt bottom. While sediment data was unavailable for the EI shelf and slope, we selected silt as sediment type which has been documented at some distance to the east [63]. The software's default settings for this sediment type were used, i.e. sediment sound velocity of $1,575 \text{ m s}^{-1}$, density of $1,700 \text{ kg m}^{-3}$ and attenuation of $1 \text{ dB per wavelength}$ [64]).“

Actually, we are not using the software's default setting (which allows basalt, silt, sandy....) as such, but the software's default setting for a silt bottom.

P9 I10: “an type Argo float” ?

Changed to “an Argo float”.

P9 I22: same as for RL, you should say more about you SL than just a foot note. You should provide some references to support the choice of SL, which do exist for fin whales.

We rephrased this paragraph, adding the requested references. Please note that the footnote is not intended to reflect on the choice of the numeric value of the source level (180 dB), but the fact that the hitherto uncommon reference value of $1\mu\text{Pa}^2\text{m}^2$ was used (commensurate with the quoted ISO norm), rather than $1\mu\text{Pa}^2\text{m}$ used commonly until now.

P9 I24: why 21m instead of 20m since you then say that you assume the whale to be singing at 20m. Also, can you provide any references form the literature that support the assumption of 20m?

We added the reference for the vocalization depth and explained that 21 m rather than 20 m is chosen by the program for computational reasons. Due to grid cells being spaced by 10.5m, as explained in the Material and Methods section.

Because you describe your mooring later, the 210m comes out of nowhere, so you should say that 210m is deployment depth.

Sentence expanded by "(i.e. the recorder's deployment depth)". This problem also becomes solved in the revised version because we moved the paragraph "Acoustic Data Acquisition" to the front of Material and Methods Section and provide the mooring information now early on .

P9 I41: Insolation? This is a weird term to use here.

Insolation describes the time integral of solar irradiance and seems hence a suitable term to describe the amount (and changes) of sunlight striking the earth's surface per day. See also https://en.wikipedia.org/wiki/Solar_irradiance.

P9 I 44 you have not yet explained what call rate is. Also, what is "call rate abundance"? Call rate is calls per unit time, but what is the abundance part?

Here the term "rate" was used mistakenly used once and we deleted it. In the revision of the manuscript the term "call rate" is now used only once for disambiguation towards "call abundance", nowhere else in the paper. See below for further details on call rate vs. call abundance.

Also, if you want to test something, you have to use statistics or explain that you will do it graphically.

We now explain that graphic comparisons are used in the Summary and Introduction

I think the methods need to have a section at the beginning that outline your general approach. This will help the reader understand the methods.

Section 3. Material and Methods section has been expanded by section 3.1 Overview on the observational techniques used. Section 3.3. Acoustic Data was restructured by adding the paragraph on fin whale calls from the original Introduction to its beginning while the leading text directly under 3.3.3 "Call Counts" has now been moved up the general introduction of section 3.3. "Passive Acoustic Data Analysis".

P10 I16: do we still think that these might be contact calls?

Yes, even in most recent papers 20-Hz pulses in irregular sequences are still referred to possibly serving a social context (establishing and maintaining contact) e.g. "Wiggins and Hildebrand (2020) Fin whale 40-Hz calling behavior studied with an acoustic tracking array. Marine Mammal Science Vol 36, Issue 3". However, we have changed the wording slightly into "...or in irregular sequences, possibly used when socializing" and added an additional reference McDonald et al 1995.

You should also briefly mention downsweeps. Why do you not consider downsweeps?

We have added information on this call type at the end of the first paragraph of "Passive Acoustic Data Analysis" and explained why we refrained from further analyses.

P10 I21: are you in the 89 o 99Hz region?

*In response to this and other comments revolving around the 85, 89 and 99 Hz HF-components we did a numerical calculation of the frequency observed in our data. To this end, we added the following sentence: “A simultaneously occurring higher frequency HF-component in the 85-89 Hz range occurred also regularly in our recordings, averaging at 85.6 ± 1.5 Hz (± 3 dB band width) for the 10-min spectrogram partly shown in **Error! Reference source not found.**, and 86 ± 4 Hz (± 3 dB band width again) when averaging over the full LTS-AU (featuring a 2 Hz resolution only).”*

P11 I8: I don't think it is relevant to say how you collected your mooring elements, aside from the fact that these may be sources of noise, and if think that this is a problem for your later analysis, then state it here clearly. Otherwise delete.

We moved this information into the Figure Caption of the mooring in the Supplement (Figure S1) added additional information that tidal flow is causing mooring induced noise in paragraph Passive Acoustic data analysis (3.3.1 Long-term spectrograms (LTS_AU)).

N.B. for reviewer: In our currently deployed mooring off Elephant Island we have replaced all metal shackles by rope shackles in an attempt to reduce this mooring induced noise.

P11 I10: why repeat the ADCP information here?

This became resolved by the restructuring of the whole manuscript and removing the duplication re. frequency and sampling scheme.

P12 I52: “perusal” is an odd choice of word

We replaced perusal by examination.

Table 2: hard to read and text cut-offs are awkward.

We transposed the table and formatted it for better readability. The referred to Table 2 is now Table 3 in the revised manuscript.

P13 I8: your clarification on abundance vs rate should not go as a footnote. Also, your footnote is confusing. To me, abundance should be the total number of calls detected, and call rate is that number over time. Please improve your explanation.

We moved the footnote to the text right after the first use of the term call abundance (last paragraph of 3.3) and expanded it: “Likewise, we explicitly seek to distinguish the term „call abundance“, which we here use to represent the number of calls discernible in a recording from the term „call rate“, which commonly is understood as the frequency an animal repeats a certain call. Call rates and call abundances are linked in a non-trivial way by number of animals calling, sound propagation and recorder characteristics. Hence call rate is an emissive attribute concerning a single individual (emitted calls per time unit), whereas call abundance is an immission metric, concerning the entire audible population (recorded calls per time unit).”

P13 I24: you start with LTSAs but the list of metrics in the previous section is in another order.

Restructured, such that order is maintained.

Figure 3: Include units in the captions and acronym complete names.

Included as suggested.

I suggest you remove the red lines, they are distracting and I would let the reader see for themselves.

Removed as suggested.

I suggest you put DAO in the panel where you report the fin index and not with sea ice cover.

We attempted this suggestion, however, the resulting figure appears overloaded with information, which is why we prefer to maintain the current composition. With DOA and FIN plots being right on top of each other and aligned in time, a comparison between the two seems easy nevertheless.

Also, you should reorganize the panels and put LTSA next to the fin index, and your environmental data next to each other.

Moving the LTSA (ex panel d) next to FIN (ex panel b) would result in a less clear temporal sequence of events as indicated by the red lines of the original plot. The chronological order of events (MVBS-peak → FIN-peak → ice-peak) seems, at least to us, best recognizable when keeping this order. However, we moved the LTSA from the bottom to the top, so the order is now commensurate with the listing and order of subchapters.

P15 I10: I still don't understand why you are looking at a 14-day running average.

Please see our answer to your comment for P07 I10 .

Section 3.3.3: Call counts is not an accurate title for this section. Please define difference between call counts and call abundance.

The term "call count" was deleted altogether from the manuscript. (We originally meant it to refer to the activity of counting calls, rather than a metric alternative to "call abundance"). In response to the comments given by reviewer 1, subsections 3.3.3.1 and 3.3.3.2 have been promoted to 3.3.3 and 3.3.4 while the leading text directly under 3.3.3 "Call Counts" has now been moved up the general introduction of section 3.3. "Passive Acoustic Data Analysis".

P15 I37 – p16 I15: this section should be tightened up or eliminated.

This section has been moved to the reworked introductory part of chapter 3.3. Passive Acoustic Data Analysis, where, we believe, it falls better in line with the overall flow of the text. The statements made therein form an integral part of the description of methods as requested by the reviewers (i.e. in the general comments of reviewer 2), as it reflects on the reliability of the acoustic metrics used. Even after thorough inspection we do not see how to possibly shorten (or even delete) it without omitting relevant methodological information.

3.3.3.2: how did you visualize spectrograms? There are no details on this here.

Indeed an oversight. We added the following text to (new numbering) 3.3.4: "Using acoustic data decimated to 500 Hz, spectrograms were calculated using RAVEN 1.4 (Hanning-window, FFT 334 points, overlap 90 %, frequency resolution 0.75 Hz, time resolution 1.33 s, displayed frequency range (linear) 0 – 125 Hz). Screening for 20-Hz pulses was performed audiovisual, "applying a subsampling scheme as suggested in Harris et al. [49] to manage temporal constraints.

4.1.1 The ADCP results section seems a bit superficial to me. Do you see diel vertical migration? This would be interesting to know since you are also looking at diel calling, I assume because you are interesting in the potential relationship between calling and DMV of prey.

We have analyzed the available ADCP data for diel vertical migration patterns of possible prey species and added this information to the text with a corresponding figure to the Supplement (Figure S5). We abstained from adding the figure to the manuscript for several reasons: 1) to avoid making it longer, 2) as this is a manuscript on fin whales, rather than krill behavior, and 3) because the diel vertical migration data is only very coarsely resolved as a result of it having been collected opportunistically for this application (The ADCP was set up was optimized for current velocity measurements, rather than MVBS).

Table 3: where is Table 3 referenced in the text???

Table 3 is now Table 5 and is referred to in 4.2.1 Acoustic Backscatter and the Discussion part 5.2.

4.1.3. Rather speculative and bit sounds more like discussion than results.

It was not immediately evident to us, to what the reviewers refers to specifically in his/her statement, so we inspected the section sentence by sentence.

Paragraph 1, sentences 1-3 give facts, with 3 supported by citation.

Paragraph 1, sentence 4: 1st half give facts, second half reflects common thinking.

Paragraph 2, sentence 1-2 are true statements (due to use of conjunctive might.)

While rightly, the text goes a bit beyond a mere presentation of results, it give true statements and puts the finding into context with the Acoustic Backscatter data presented before, which is why we found it to fit well here.

P19 I20 what do you mean by anisotropic?

Anisotropic describes the fact that the detection range varies with direction.

4.1.4 This entire section describes results in a supplementary figure which makes me think that the figure should not be supplementary.

Now. 4.2.4. Figure added to main manuscript

Also, "sun's yellow" vs "bright yellow"?

We are not quite sure what the reviewer refers to with this comment, it might be a printer issue, but we changed naming to orange vs. yellow.

P19 I44: you never mention in the methods that you use sound speed profiles from summer and winter. You only mention January CTD measurements.....

Information added to the methods section: "taken by an Argo float (AWI 0246, WMO 7900409) in summer on 16 Jan 2013 at 60°02.142'S 57°29.25'W, and in winter on 12 Oct 2013 at 57°50.658'S 39 13.53'W,..."

4.2.3: what is FINxx? You also use FINs and FIN index in the text. Please make sure you fully define all your acronyms. I assume you are referring to fin index but please be consistent and spell everything out.

(now 4.1.2.) xx was meant to be a placeholder for 21 (Hz) or 89 (Hz). As this does not seem to be immediately evident, we replaced it by the FIN21-AU and FIN89-AU throughout the manuscript as applicable.

N.B. FIN89 became FIN86 in the new version.

4.2.4 The first paragraph is really more discussion than results.

(now 4.1.4) This paragraph gives the motivation for conducting and presenting the call abundance screening by briefly listing the possibilities that might have caused the observed patterns in the FIN21-AU and FIN-89-AU indices. We find this helpful and do not see the mere listing of options as a discussion.

4.2.4.2: What was your sample size for diel analysis? Same as previous section?

Yes, as indicated by “SonoVault call abundances (CAB-SV) between January and November 2013 seem not to exhibit a diel cycle in relation to the local light regime (Figure 6).”

Figure 5: Time unit is weeks?

*Time tics advance by 6-days and represent the center of the 24 h time period analysed (see section 3.3.4 (ex 3.3.3.1) for details) This format was chosen (and is maintained) as each tic represents an analysed day (i.e. **the effort**) even if the count was 0, like on the days in August-Sept-October. This “effort” information would get lost in the “monthly” tics commonly used.*

To preserve this information, we decide to maintain the x-axis labels in its current form, while accommodating the reviewers’ comments by adding a standard time axis on the top of the figure, along with color shading for the different seasons.

The caption, units and axis title are confusing. These are weekly average fin whale calls per 10-min files?

We re-rendered the figure and the caption was rephrased. Regarding time tics, see comment above. The boxplot represents “the total number of 20-Hz pulses within a 10-min file ... from every 7th 10-min-file of every 6th day.” i.e. one daily box every 6th day, as described in the first paragraph of section 3.3.4

Also, this caption is the first mention that files are discarded due to background noise.

This information is given in section 3.3.4 (ex 3.3.3.1) “Of the available files, a total of 219 files (mainly August, September and October ...) had to be excluded from the analysis due to excessive ambient noise.”

P23 l19: why the list of percentages? Can you not plot this?

Replaced by “20-Hz pulses were present for 9 of 11 months in the SonoVault data (CAB-SV): January through August but not in September and November, with December not sampled.”, as percentages were not essential to discussion.

P23 I 32 here is the first mention of statistics! You should have mentioned this in methods!

We have added information on statistical analysis in Section 3.3.4 (new), at the end of the paragraph: “Diel calling pattern analysis was performed on CAB-SV data from March-July, as these are the months with high 20-Hz pulse detections and a clear day/night cycle. To test for statistically significant differences in call activity in dependence on light period a Kruskal-Wallis test (as a Kolmogorov-Smirnov test showed data not to be normally distributed, $p < 0.001$) was applied and differences were accepted as statistically significant, when $\alpha < 0.05$.”

P25 I58 “Regular 20-Hz call sequences are thought to be related to breeding contexts, serving mate attraction or advertisement of resources” is irrelevant in this section. Why are you discussing the function of calls here? And I think you have already mentioned this a few times in the text.

Reduced to once mentioned in Passive Acoustic Data Analysis, and once in Confounding Factors of Discussion.

Here you mention the 40-Hz downsweeps for the first time (I think you should mention them in the intro as well).

Added to section 3.3 Passive Acoustic Data Analysis: “Fin whales also produce frequency modulated downsweeps, which generally range between 100 Hz and 30 Hz and are often referred to as 40-Hz {Watkins, 1981 #149; Širović, 2013 #4216} calls. This call has been attributed to calling during social interaction and foraging and is less well described the 20-Hz pulse.”

What other species do they get confused with? I think you should be able to distinguish them from blue whale Dcalls because they are shorter.

Others species producing similar downsweeps are Antarctic blue whales (see Rankin et al 2005) Antarctic minke whales (see Dominello and Sirovic 2016), sei whales (Calderan et al 2014) and pygmy blue whales (Gavrilov et al 2012), so a reliable discrimination whether the Fm-calls were produced by fin whales was not possible due to similar spectral characteristics. Antarctic blue whales for instance were acoustically present during most of the recording time, suggesting that FM-calls are at least partly produced by blue whales which then may obscure a potential calling pattern of fin whale Fm-calls. Additionally, flow related mooring noise appeared to cause like noises, as easily identified by its daily and fortnightly patterns matching the tidal cycles.

Manuscript expanded accordingly, yet in a more compact form under Confounding factors in the discussion -> 5.6.2.

P26 I49: this footnote should be in the results and should be discussed here.

This statement was deleted from the new more focussed version, as it did not contribute to the topics discussed therein.

P27 I 12: ref 60 does not refer to northern Chile and neither reference compares seasonality in sightings and acoustic presence.

Correct. However, Buchan et al. 2019 provide the following text, which is what we refer to: (p 143)

“From recent work off northern Chile (S.J. Buchan et al. unpubl. data) high visual sightings of fin whales during spring and early summer month (November-February) do not coincide with acoustic recordings of fin whale vocalizations, suggesting variation in vocalization production over time”.

We contacted Susannah Buchan regarding this statement and she confirmed a lag between sightings data in summer and song recorded in winter, with the additional remark that sighting effort is much reduced during winter moth.

The second Reference [76] was cited falsely, it should have been Širović, A., et al. 2013 Temporal separation of two fin whale call types across the eastern North Pacific. Marine Biology. 160, 47-57. (<https://doi.org/10.1007/s00227-012-2061-z>), where the authors state in their introduction “It is interesting, however, that peak acoustic presence based on 20-Hz calls off Southern California lags the peak in presence from visual survey and occurs in the fall and winter (Clark and Fristrup 1997; Oleson 2005).” We have changed the reference accordingly.

We changed the wording of the sentence to “. Passive acoustic data collected in this study clearly lags the peak reported in visual sightings (e.g. [13]), similar to observations from Southern California and northern Chile, “where the peak in 20-Hz song presence also lags visual sightings” (Širović, 2013 #4216) and S. Buchan, pers. communication).”

P27 I17: where do you consult visual sighting data in this study?

We refer to visual sightings in the Introduction with additional information in Supplement Table S1 and the Discussion parts 5.1, 5.5, 5.6.2. We attached an additional figure to the supplement with available sightings since 2000 around our recording site (Supplement Table S8). The restructuring of the manuscript should make the approach of comparing acoustic data with visual sighting now clearer.

P27 I46: rewrite “yet nevertheless it is”.

Changed accordingly.

P27 I51: what do you mean by “traditional” in this context?

The term “traditional” was intended to describe locations that used to be ecologically important in the past (i.e. pre whaling) but are (ecologically) not anymore, yet are still frequented because of cultural or inherited behavioural patterns.

P28 I30 “supports the suggestion that call production is related more exclusively to a specific behavioural context rather than to environmental parameters”. This is a curious statement. I would say that call production is ALWAYS related to (acoustic) behaviour. Maybe what you are trying to say is that call rates here are responding to changes in call production/acoustic behaviour rather than changes in animal presence driven by environmental correlates. Is this what you mean?

Yes, your interpretation is correct and was considered when rephrasing the manuscript.

Table 4: There is quite a lot of information lacking from the caption and this makes for difficult reading of this table. What is the last column “daily averaged FIN”? You also have to say in the text why you think the other sites in the table are relevant here, especially the mid-latitude site (Chile HA03), given that the calls do look different (peak frequencies at 17Hz and 85Hz).

We expanded the caption to include all abbreviations. Thank you for pointing towards the potential implications of the 85Hz vs. 89 Hz difference. See next response for details.

P29 I 60: why you think the mid-latitude site (Chile HA03) is relevant here given that the calls are at different frequencies (peak frequencies at 17Hz and 85Hz)? From the text it looks like you assume that they are the same acoustic population.

Thank you for pointing towards the potential implications of the 85 Hz vs. 89 Hz difference. We went back to the data and calculated our peak frequency, which turned out to be 85.7 Hz, making the assumption of the EI and JFA groups belonging to the same population even more convincing, which is why we explored this aspect and discussed it in more detail in the revision.

5.2.1 and 5.2.2: I strongly suggest you tighten up this text and put more emphasis on the interesting environmental data you have here, rather than too much discussion on context of calls and super groups. I also suggest you rethink the titles of these sections, so they better reflect the points that you are making. I also think that your discussion on the function of calls should be linked to the fact that you see no diel patterns, indicating that these calls are not responding to DVM of prey and herefore likely not linked to feeding.

Basically implemented as suggested, see 3 comments farther down for our detailed response. DVM of prey is now included in Methods 3.4.1, Results 4.2.1 and the Discussion. We also added a figure to the Supplement Figure S5.

P29 I38: I think this paragraph is a bit speculative and could be reduced or eliminated.

Paragraph deleted.

Same for the discussion about super groups that follows. I would strongly suggest you reduce this.

We deleted all instances where super groups were discussed.

5.2.4 and 5.2.5 I would suggest these sections be expanded which 5.2.1 and 5.2.2 reduced.

The entire discussion was rewritten. It now starts with the ecological aspects (Sections 5.1 to 5.5) and ends with the confounding factors 5.6. Section 5.1 to 5.5 have been rearranged according to reviewer's suggestion.

5.2.4 old was moved and expanded in 5.3 "Where do they go, where do they come from"

5.2.5 old was moved into 5.4 "Does sea ice affect physical presence in the EI area". Datavise this was not expanded, e.g. by a statistical test, as it is obvious from the progression of the data that sea ice does not affect fin whale presence. However we provided further support to our observation by listing similar findings reported in the literature.

5.2.1 old was moved into 5.1 "When do fin whales use the EI area" and consolidated (328 words new vs. 414 words old)

5.2.2. old was moved into 5.2 "What drives their acoustic display in the EI area?"

The section 5.6.2 Confounding Factors – Acoustic occurrence vs. physical presence grew somewhat due to an in-depth consideration of the potential biases introduced by focusing on the 20-Hz pulse in our analyses.

P 35 I43. "Summarizing, we note that while CAB and FIN indicated a 10-fold increase over the past 10

years, confounding factors prohibit assigning this unequivocally to a like increase of population.” Yes, you are right that using acoustics to look at population trends is a critical area of research right now. This whole section is really interesting but I think you need to take it out of the discussion and put it in the results and present clearly in the methods how you are calibrating CAB and FIN and comparing results from all three previous studies with your own. I think this would be a very valuable addition to the current knowledge if you can do this.

See earlier response to this aspect in section general comments: “We moved the subclauses providing concrete values of FIN and CAB as derived by our study to the respective Results section. The methodology of calculating these values is described in the Material and Methods sections. No additional calibration of CAB and FIN is performed. The only formula used here to check the trends in CAB and FIN for plausibility is the known relation between sound intensity and sound pressure level, which does not require extra mentioning in the Materials and Methods section. We consider comparisons with data from other studies as well as evaluating the plausibility of changes in FIN versus changes in CAB to be well placed in the discussion, which is why we kept those parts there.”

Conclusions: this need to be more concise and really just focus on your main conclusions and brief implications rather than too much more discussion about Antarctic krill.

The conclusion was reworked in line with the restructuring of the Discussion section.